# Robots Pre-train Robots: Manipulation-Centric Robotic Representation from Large-Scale Robot Datasets

**Guangqi Jiang**[1][*]   **Yifei Sun**[2][*]   **Tao Huang**[3][*]   **Huanyu Li**[3]   **Yongyuan Liang**[4][†]   **Huazhe Xu**[5][†]

[1] University of California, San Diego   [2] Tongji University   [3] Shanghai Jiao Tong University
[4] University of Maryland, College Park   [5] Tsinghua University

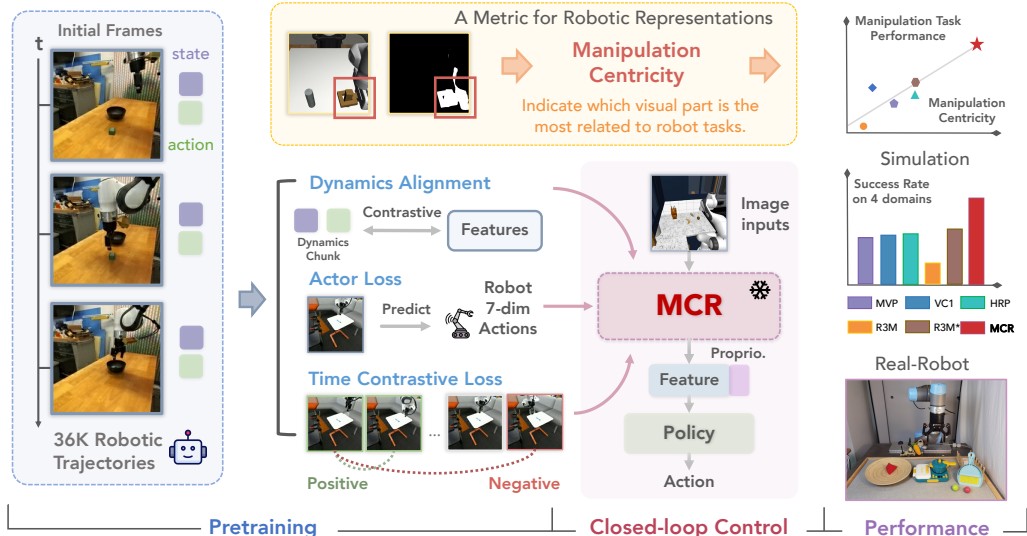

**Figure 1: Overview.** We introduce a robotic representation evaluation metric termed *manipulation centricity*, which exhibits a strong correlation with downstream policy performance. Accordingly, we design a new pre-training method, **MCR**, to learn manipulation-centric representation from large-scale robotic datasets. Comprehensive experiments on both simulations and real robot validate the superiority of our proposed representation.

## Abstract

The pre-training of visual representations has enhanced the efficiency of robot learning. Due to the lack of large-scale in-domain robotic datasets, prior works utilize in-the-wild human videos to pre-train robotic visual representation. Despite their promising results, representations from human videos are inevitably subject to distribution shifts and lack the dynamics information crucial for task completion. We first evaluate various pre-trained representations in terms of their correlation to the downstream robotic manipulation tasks (i.e., manipulation centricity). Interestingly, we find that the "manipulation centricity" is a strong indicator of success rates when applied to downstream tasks. Drawing from these findings, we propose **M**anipulation **C**entric **R**epresentation (**MCR**), a foundation representation learning framework capturing both visual features and the dynamics information such as actions and proprioceptions of manipulation tasks to improve manipulation centricity. Specifically, we pre-train a visual encoder on the DROID (Khazatsky et al., 2024) robotic dataset and leverage motion-relevant data such as robot proprioceptive states and actions. We introduce a novel contrastive loss that aligns visual observations with the robot's proprioceptive state-action dynamics, combined with an action prediction loss and a time contrastive loss during pre-training. Empirical results across four simulation domains with 20 robotic manipulation tasks demonstrate that MCR outperforms the strongest baseline by 14.8%. Additionally, MCR significantly boosts the success rate in three real-world manipulation tasks by 76.9%. Project website: robots-pretrain-robots.github.io.

---

[*]Equal contribution. [†]Equal advising. Correspondence to `gqjiang@ucsd.edu`.

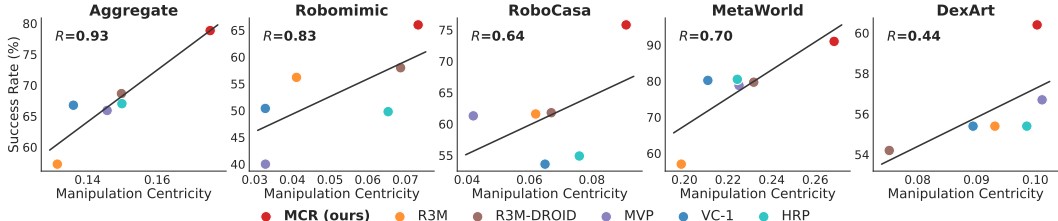

**Figure 2: Correlation between manipulation centricity and downstream performance.** Our findings reveal that (1) the proposed metric of manipulation centricity strongly correlates with the downstream performance of robotic representations, and (2) using the robot dataset DROID yields greater benefits for robotic representations than human datasets. (3) These insights motivate our method, MCR, which leverages dynamics labels from the robot dataset to further enhance manipulation centricity and downstream performance.

# 1   INTRODUCTION

Grounding robots with generalizable visual representations, termed robotic representation, is crucial for real-world visuomotor control. Pre-training robotic representations on extensive in-domain data offers a promising strategy for robotics, drawing from the success of large-scale pre-training in computer vision (He et al., 2022) and natural language processing (Devlin et al., 2019). However, due to the scarcity of robotics data and high collection costs, recent studies have utilized everyday human operation videos (Grauman et al., 2021; Goyal et al., 2017) to pre-train visual representations for robotic manipulation, positing that knowledge behind human manipulation can inform representations in question. Various levels of human manipulation knowledge — such as task semantics (Nair et al., 2022), pixel-level comprehension (Xiao et al., 2022; Majumdar et al., 2023), and physical interaction (Jia et al., 2024; Srirama et al., 2024)— have demonstrated their effectiveness in benefiting robotic representation. A fundamental question naturally arises: **(Q1) What specific features captured from human data significantly contribute to improving robotic manipulation?**

To investigate this question, we conduct a comprehensive evaluation of prior representations concerning their policy learning performance across various downstream simulation domains. Interestingly, we observe a correlation between a representation's downstream performance and its ability to capture manipulation-relevant regions including robot end-effectors and task-relevant objects. To examine this correlation more formally, we introduce a metric, 'manipulation centricity', and propose an evaluation protocol to quantify it across different representations. The core of this protocol is to measure the similarity between ground truth manipulation regions and the focus of the representation. Our results indicate a strong correlation between downstream performance and manipulation centricity, as illustrated in Figure 2. These findings provide valuable insights into our first question: **(A1) Manipulation centricity emerges as the key factor contributing to enhanced robotic manipulation performance.**

However, human videos not only introduce inherent distribution shifts due to the human-robot embodiment gap, but also lack the dynamics information essential for successful task execution. This naturally leads to the second question: **(Q2) Is there a better dataset choice than human dataset to learn manipulation centricity in robotic representation?** With the recent emergence of large-scale robot datasets (Collaboration et al., 2024; Walke et al., 2023), we hypothesize that the smaller domain gap presented by these datasets may naturally be more suitable for learning manipulation centricity. To prove this hypothesis, we re-train the prior method with a representative robot dataset, DROID (Khazatsky et al., 2024), and indeed observe significant improvements in both performance and manipulation centricity. This answers our second question: **(A2) Large-scale robot datasets can be a better choice than human datasets for learning manipulation centricity.**

Recognizing that robotic datasets provide more relevant information about robot embodiment trajectories, this raises the third question: **(Q3) How to learn manipulation centricity better with large-scale robot datasets?** Our starting point is the dynamics labels, including robot proprioceptive states and actions, in the robot dataset but absent in the human dataset. We consider that these dynamics labels contain the core knowledge behind accomplishing a manipulation task, which has not been explicitly utilized in previous pre-trained robotic representations from human data. To this end, we introduce a new method, **M**anipulation **C**entric **R**epresentation (**MCR**), designed to leverage dynamics labels from robot dataset to improve manipulation centricity of robotic representation.

Specifically, we propose two training objectives: dynamics alignment loss, aligning pixel representations with robot state-action pairs at the same timestep, and action prediction loss, predicting robot actions from image observations. We also incorporate a time contrastive learning objective (Nair et al., 2022) to encode temporal information. Integrating these objectives into the training process significantly improves manipulation centricity, leading to a 14.8% performance increase across four simulation domains encompassing 20 diverse robotic tasks, and a 76.9% improvement across three real-world robot tasks. This answers our third question: **(A3) Effectively utilizing dynamic labels enhances the learning of manipulation centricity.**

We summarize this paper in Figure 1, highlighting three key insights: (1) Manipulation centricity serves as an indicator of the effectiveness of representations for robotic control, as evidenced by the strong correlation with downstream performance. (2) Large-scale robot datasets can be a superior choice compared to human datasets for learning manipulation centricity, as indicated by significant improvements in performance and manipulation centricity when using robot datasets. (3) Effective utilization of dynamics labels from robot datasets significantly enhances the learning of manipulation centricity in robotic representations, demonstrated via the introduction of the MCR method, which incorporates the proposed dynamics alignment and action prediction objectives.

## 2 EXPERIMENTAL SETUP: EVALUATING ROBOTIC REPRESENTATIONS

This section outlines the experimental setup used to evaluate the effectiveness of pre-trained visual representations for robotic manipulation. In line with prior work (Nair et al., 2022; Xiao et al., 2022), we freeze the pre-trained encoders and utilize Imitation Learning (IL; Argall et al. (2009)) for downstream policy learning. Accordingly, the quality of the visual representations is assessed based on IL performance averaged across various downstream tasks.

**Evaluation protocol.** To assess a visual encoder $\mathcal{F}_\phi$, which maps an RGB image $I \in \mathbb{R}^{H \times W \times D}$ to a continuous feature vector $z = \mathcal{F}_\phi(I)$, we introduce a policy network $\pi_\theta$ built atop the frozen encoder $\mathcal{F}_\phi$. This policy network takes as input the feature vector $z$ and the robot's proprioceptive state $s$, outputting an action distribution $\hat{a} \sim \pi_\theta(\cdot|z, s)$. To train the policy, a task-specific demonstration dataset $\mathcal{D}_{IL} = \{\tau_1, \tau_2, ..., \tau_n\}$ is collected, where each demonstration $\tau_i$ is a trajectory consisting of a sequence of expert-level observation-action pairs $\tau_i = \{(I_t, s_t, a_t)\}_{t=1}^T$. Then, the policy is optimized via the Behavior Cloning (BC; Bain & Sammut (1995)) algorithm, where the optimization objective is to minimize the error between the predicted action $\hat{a}$ and the optimal action $a$ from demonstration data. For a fair comparison, we employ the same BC algorithm for any used pre-trained encoder in each task. During evaluation, the policy is executed in a closed-loop manner within the environment with online feedback to test its success rate on the target manipulation task. We evaluate at least 20 episodes every fixed number of training epochs, selecting the highest success rate. The mean and standard error of success rates across three seeds are reported.

**Simulation environments and datasets.** We select a total of 20 tasks across 4 simulation environments to evaluate representation quality. These tasks encompass a range of end-effectors, including grippers and dexterous hands, along with diverse robot arms and varying levels of manipulation complexity. Task visualizations are shown in Figure 3, with additional details provided in Appendix C.1.

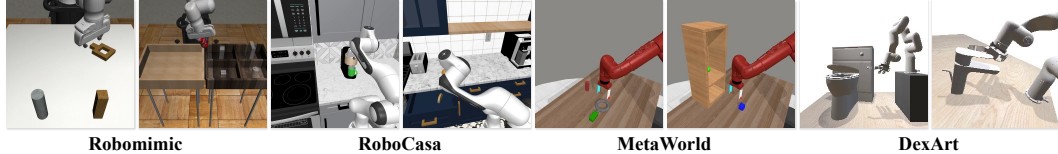

**Figure 3: Task visualization.** We consider 20 challenging and diverse manipulation tasks spanning 4 domains.

- Robomimic (Mandlekar et al., 2021) features a benchmark suite for tabletop manipulation using the Franka Emika Panda arm and a parallel gripper. We select 3 challenging tasks, utilizing 200 demonstrations from the official proficient human teleoperation dataset for each task.

- RoboCasa (Nasiriany et al., 2024) is a realistic simulation platform focusing on housing scenarios with a Franka Emika Panda arm. We select 3 challenging tasks covering different kitchen scenarios and utilize 50 demonstrations for each task following their official training scripts.

**Table 1: Grad-CAM visualization.** We present Grad-CAM visualizations alongside their corresponding task success rates for the Square task from Robomimic and the Pick Place Wall task from MetaWorld. Representations that capture the robot's end-effectors and task-relevant objects are linked to improved downstream performance. Comprehensive visualizations can be found in Appendix D.

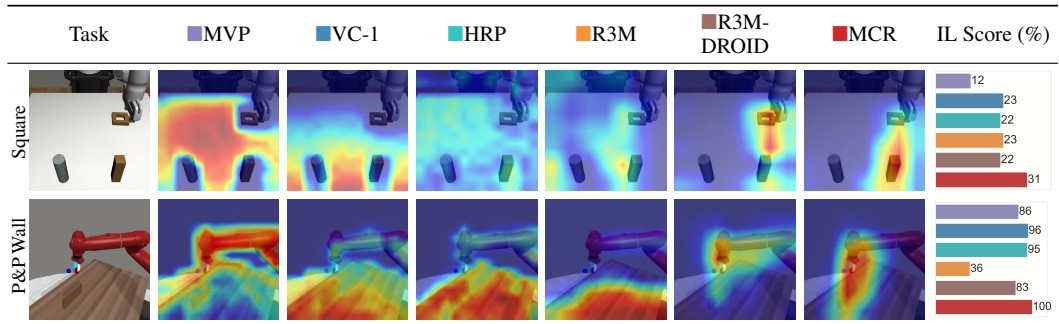

- MetaWorld (Yu et al., 2019) features a benchmark suite of robotic tabletop manipulation with the Sawyer robotic arm and gripper. We select 10 diverse tasks according to both the difficulty categorized in Seo et al. (2022) and the evaluation suite in Nair et al. (2022), and collect 25 demonstrations via official scripted controllers for each task.

- DexArt (Bao et al., 2023) features a benchmark suite for dexterous manipulation tasks on articulated objects using the XArm6 arm and an Allegro hand. We select all 4 tasks and gather 100 demonstrations rollout by well-trained RL policies for each task.

**Comparison methods.** We consider representative methods in robotic visual representation:

- MVP (Xiao et al., 2022) pre-trains a Vision Transformer (ViT, Dosovitskiy et al. (2021)) using Masked Auto-Encoding (MAE, (He et al., 2022)) on a mix of human-object interaction videos.

- VC-1 (Majumdar et al., 2023) is trained similarly to MVP but also additionally incorporates navigation and the ImageNet (Deng et al., 2009) dataset during pre-training.

- HRP (Srirama et al., 2024) fine-tunes a pre-trained ViT to predict human affordance labels, including hand pose, contact points, and active objects extracted from human videos.

- R3M (Nair et al., 2022) pre-trains a ResNet (He et al., 2015) model on human videos using time contrastive learning and video-language alignment to extract temporal and semantic information.

## 3 INVESTIGATION: MANIPULATION CENTRICITY OF REPRESENTATIONS

Prior work utilizes in-the-wild human videos to assist robotic representation learning, yet the domain gap between human and robotic tasks may influence the representation quality. To investigate this problem, we take an initial step by visualizing these representations in simulated tasks. Interestingly, we observe that a representation's downstream task performance appears to correlate with its ability to capture manipulation-relevant regions. To investigate this correlation more formally, we introduce a metric, 'manipulation centricity', and propose an evaluation protocol to measure it across different methods. Our results indicate a strong correlation between manipulation centricity and downstream task performance, which strongly guides our method design in the next section. We introduce the whole pipeline in the following parts and present more details in Appendix D.

**Feature visualization & motivation.** We use Gradient-weighted Class Activation Mapping (Grad-CAM; Selvaraju et al. (2017)), a widely-adopted network visualization technique in computer vision, to analyze the features of existing representations. Grad-CAM highlights the regions of an input image that are most influential in the model's decision-making process. In our context, this technique reveals how the network interprets images within manipulation tasks, as shown in Table 1. Specifically, in the Square task, we observe that the MVP model tends to focus on irrelevant regions, such as the table, rather than the manipulator or objects, while in the Pick Place Wall task, R3M exhibits a similar pattern. Both cases correspond to poor downstream performance. In contrast, representations that emphasize the robot's end-effectors and objects are associated with better downstream performance. This motivates our investigation into whether representation quality is linked to its ability to capture these 'manipulation-relevant' regions, a property we define as *manipulation centricity*.

**Measuring manipulation centricity.** To quantify manipulation centricity, we compute the similarity between the regions highlighted by Grad-CAM and the ground truth regions corresponding to the end-effector and task-relevant objects. The ground truth is generated using the powerful video segmentation model SAM 2 (Ravi et al., 2024), with manual annotation of key points within the target regions. This approach enables efficient annotation across an entire video of task execution. We apply this annotation method to create segmentation masks for demonstration data $D_{IL}$ across all simulated tasks, forming our evaluation dataset. As illustrated in Figure 4, manipulation centricity is then measured by averaging the Jaccard Index, a widely-used similarity metric in image segmentation, between the binarized Grad-CAM outputs and the ground truth segmentation masks over the entire evaluation dataset. All the ground truth annotations are available in Appendix D.

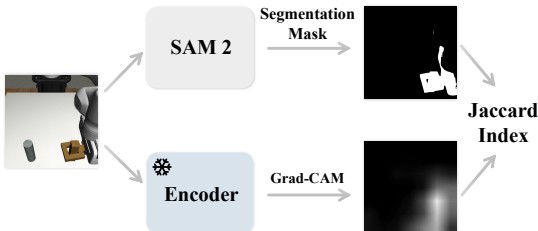

**Figure 4:** Measurement of manipulation centricity.

**Key findings.** The aggregate results, presented in Figure 2, reveal a strong correlation between manipulation centricity and downstream task performance, with a Pearson correlation coefficient of $R = 0.93$. We also evaluate this correlation within individual simulation domains, where the evaluation dataset consists of demonstrations specific to each domain. The positive correlation remains consistent across all domains, albeit with varying strengths. In summary, these results suggest that manipulation centricity measured on our entire evaluation dataset is a reliable indicator of the effectiveness of robotic representations.

## 4 MCR: LEARNING MANIPULATION-CENTRIC REPRESENTATION

Drawing from the conclusions in Section 3, our focus shifts to improving the manipulation centricity of robotic representations in this section. This is achieved by two attempts in our proposed method, **M**anipulation **C**entric **R**epresentation (**MCR**). First, we find that re-training existing models with large-scale robot data (*i.e.*, DROID (Khazatsky et al., 2024)) instead of human data can significantly improve manipulation centricity. This indicates the value of utilizing robot-specific datasets, as discussed in Section 4.1. Second, we introduce novel pre-training objectives that leverage the robot's state-action dynamics—information provided in robot data, which is inherently absent in human datasets, to further enhance manipulation centricity, detailed in Section 4.2.

### 4.1 LARGE-SCALE ROBOT DATASET

**Dataset processing.** In recent years, several large-scale robot datasets have been introduced (Brohan et al., 2023; Walke et al., 2023; Collaboration et al., 2024; Khazatsky et al., 2024). Among these, we select the DROID dataset for our method because of its extensive scene diversity and a large volume of data. The dataset is collected using the Franka robot arm and Robotiq 2F-85 gripper via teleoperation, comprising a total of 76k trajectories. Each trajectory includes RGB images from two external Zed 2 cameras, robot proprioceptive states, and actions consisting of delta 6D poses and 1-DoF gripper actions. To ensure data quality, we filter out trajectories with fewer than 40 timesteps to ensure adequate temporal information in each trajectory. Additionally, trajectories containing incomplete or single-word language instructions are removed, as these may indicate lower-quality interactions. After processing, we retain 36k trajectories for pre-training.

**A motivating discovery.** Many previous methods rely on human-object interaction datasets to learn manipulation concepts from human behavior (Nair et al., 2022; Xiao et al., 2022; Srirama et al., 2024). However, the domain gap between human hands and robotic manipulators may inherently affect representation quality. We re-train the R3M model using the robot-specific DROID dataset, yielding a variant we call R3M-DROID (equivalent to R3M*). As expected, Grad-CAM visualizations in Table 1 show that R3M-DROID better captures manipulation-relevant regions, and quantitative results downstream performance, as shown in Figure 2, confirms this improvement. This suggests that robot-specific data inherently benefits representation learning by narrowing the domain gap between training and deployment environments.

## 4.2 TRAINING MCR

Unlike human datasets, robot datasets provide access to manipulator dynamics, including proprioceptive states and actions. While previous approaches do not fully leverage this dynamics information, we hypothesize that incorporating it will enhance manipulation centricity. To achieve this, we design two new training objectives that explicitly leverage robot dynamics. Additionally, we adopt a semantic learning loss from prior work (Nair et al., 2022) to retain semantic information in the representations. These objectives and their integration into our training process are detailed below.

**Dynamics alignment.** Our first insight is that each image observation corresponds to an underlying proprioceptive robot state at every timestep. We aim to learn this correspondence, termed dynamics alignment, through contrastive learning. Formally, we define a state-action dynamic chunk of length $l$ at timestep $t$ as $d_t = [s_{\lceil t - \frac{l}{2} \rceil}, a_{\lceil t - \frac{l}{2} \rceil}, s_{\lceil t - \frac{l}{2} + 1 \rceil}, \ldots, s_{\lfloor t + \frac{l}{2} \rfloor}]$. The positive sample for $d_t$ is its corresponding RGB image $I_t$ at the same timestep, while the negative samples are drawn from a different timestep $k$ within the same trajectory. During training, we randomly choose $t$ and $k$. The encoder $\mathcal{F}_\phi$ is trained to discard irrelevant details from high-dimensional images and retain the essential information for manipulation. We employ the InfoNCE loss (van den Oord et al., 2019) for contrastive learning, introducing an MLP projector $H$ to map the dynamics chunk $d_t$ to the same dimension as the image feature vector $z = \mathcal{F}_\phi(I)$.

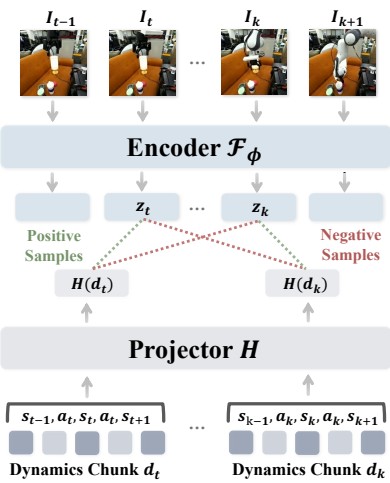

**Figure 5: Illustration of objective** $\mathcal{L}_{\mathrm{dyn}}$.

The objective function is illustrated in Figure 5 and formalized as:

$$\mathcal{L}_{\mathrm{dyn}} = -\sum_{b \in \mathcal{B}} \log \frac{e^{\mathcal{S}(z_t^b, H(d_t)^b)}}{e^{\mathcal{S}(z_t^b, H(d_t)^b)} + e^{\mathcal{S}(z_t^b, H(d_k)^b)}}, \tag{1}$$

where $\mathcal{S}$ represents the negative L2 distance, and $b$ denotes a sample from the batch $\mathcal{B}$.

**Action prediction.** We also integrate a behavior cloning (BC)-like actor into our pre-training framework, based on the idea that robotic representations should be predictive of expert-level behaviors in the dataset. The actor is implemented as a shallow MLP head that maps the image feature vector $z_t$ to the predicted robot actions $\hat{a}_t$. We use mean squared error as the objective for action prediction:

$$\mathcal{L}_{\mathrm{act}} = -\sum_{b \in \mathcal{B}} \mathrm{MSE}(a_t^b, \hat{a}_t^b). \tag{2}$$

**Temporal contrast.** We also wish the representation to encode temporal information, which has shown importance for manipulation tasks (Zhao et al., 2023). To this end, we adopt the time-contrastive learning objective from Nair et al. (2022), which encourages temporally close frames in a video to be closer in the embedding space than those that are temporally distant or from different videos. For this, we sample a frame triplet $(I_u, I_v, I_w)$ where $u < v < w$, and compute the following loss:

$$\mathcal{L}_{\mathrm{tcl}} = -\sum_{b \in \mathcal{B}} \log \frac{e^{\mathcal{S}(z_u^b, z_v^b)}}{e^{\mathcal{S}(z_u^b, z_v^b)} + e^{\mathcal{S}(z_u^b, z_w^b)} + e^{\mathcal{S}(z_u^b, z_u^{\neq b})}} \tag{3}$$

where $z_u^{\neq b}$ is a negative sample from a different video within the batch $\mathcal{B}$.

**Oveall objective & implementations.** We train MCR with a combination of introduced objectives:

$$\mathcal{L}_{\mathrm{MCR}} = \mathcal{L}_{\mathrm{dyn}} + \mathcal{L}_{\mathrm{act}} + \mathcal{L}_{\mathrm{tcl}}. \tag{4}$$

We do not introduce additional hyperparameters for weighting these objectives, as $\mathcal{L}_{\mathrm{MCR}}$ already yields strong empirical performance. Our default encoder backbone is ResNet-50, and we use Adam (Kingma, 2015) as the optimizer. The encoder is trained for 500k steps to ensure convergence, and we select the last checkpoint as the final model. The whole training process is used **50 hours with a single NVIDIA 3090**. Further training details are provided in Appendix C.

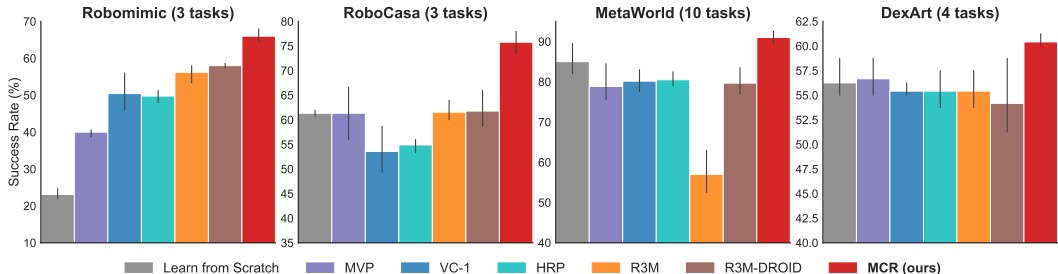

**Figure 6: Simulation results.** We evaluate MCR and baselines across different domains. Our method consistently outperforms the baselines. Results are mean success rate aggregated over 3 seeds with standard deviation.

## 5 EVALUATION AND ANALYSIS OF MCR

Our experiments are conducted in simulation and the real world to answer the following questions:
**(1)** Does MCR learn manipulation centricity and outperform baselines in simulation? (Section 5.1)
**(2)** Can the conclusions from **(1)** generalize to real-world manipulation tasks? (Section 5.2)
**(3)** Which design choices of MCR matter for learning manipulation centricity? (Section 5.3)
**(4)** What benefits does large-scale robot data bring to robotic representation learning? (Section 5.4)

### 5.1 SIMULATION RESULTS

**MCR does improve manipulation centricity.** We begin by presenting visualizations of Grad-CAM results for MCR in Table 1 (see all results in Table 8). Qualitatively, our representation excels at capturing manipulation-relevant regions of each task. For instance, in the Robomimic Square task, MCR focuses on the gripper, square tool, and the target area—key elements of the task. In contrast, other methods like VC-1 and MVP either fail to highlight these areas, or, like R3M and HRP, only capture them partially. This improved localization of task-relevant features is further confirmed quantitatively in Figure 2, where MCR significantly enhances manipulation centricity across all domains. Notably, MCR even highlights the optimal path the end-effector should follow, suggesting that our representation also learns essential information related to task execution.

**MCR outperforms baselines on simulation tasks.** Thanks to its improved manipulation centricity, MCR delivers substantial downstream performance gains compared to the strongest baseline methods across all selected domains, as visualized in Figure 6. This holds even in the DexArt domain, which involves a dexterous hand as the end-effector—a setup different from the gripper used in the DROID pre-training dataset. Moreover, in the MetaWorld domain, where prior baselines struggle and perform no better than the Learning-from-Scratch (LfS) method, a strong baseline with data augmentation implemented by Hansen et al. (2023), MCR maintains a significant advantage. We attribute the poor performance of baselines in MetaWorld to the limited number of demonstrations, which makes it difficult for policies to leverage the pre-trained representations. In contrast, MCR reduces the policy's learning burden by providing manipulation-centric features that better fit the task. In summary, our experiments suggest that pre-training with large-scale robot data, along with optimizing for manipulation centricity, yields representations that significantly improve performance in simulated manipulation tasks.

### 5.2 REAL ROBOT RESULTS

**Experimental setup.** As visualized in Figure 7, our real-world experiments involve a UR5e arm equipped with a Robotiq 2F-85 gripper and a RealSense D435i camera for RGB image capture. We designed three tabletop manipulation tasks with different objects and manipulation skills:

- Lift. The robot grips the sandbag on the plate and lifts it up in the sky.
- Sweep. The robot grasps the broom to sweep the trash from the table into the dustpan.
- Rearrange. The robot picks up the on-table pot and places it at the designated spot on the stove.

The demonstrations used for BC-based policy training are collected by keyboards with the human operator, with 30 collected for Lift and 40 for more difficult tasks Rearrange and Sweep. The pre-trained representations are frozen during policy training, inheriting the simulation setup. For a fair

**Figure 7: Real robot setup.** We design 3 real-world robot tasks with different manipulation skills and objects.

comparison, we evaluate each method with the same sets of start-up conditions, which are unseen in demonstrations, in each task. More experimental details can be found in Appendix C.5.

**MCR significantly surpasses baselines on real robot tasks.** The evaluation results, shown in Table 2, demonstrate that MCR consistently outperforms the baseline methods across all tasks. In the Lift and Rearrange tasks, baseline methods often fail to grasp the object accurately, particularly when object positions were unseen in the demonstrations. In the Sweep task, their performance deteriorates further due to poor accuracy in sweeping, with the trash frequently missing the dustpan and rolling off in unintended directions. In contrast, the policies trained with our representation exhibit robust and generalizable handling of these complex tasks.

| Task | LfS | MVP | VC-1 | R3M | **MCR** |
|------|-----|-----|------|-----|---------|
| Lift | $^5/_{10}$ | $^6/_{10}$ | $^5/_{10}$ | $^6/_{10}$ | $\mathbf{^9/_{10}}$ |
| Sweep | $^3/_{10}$ | $^1/_{10}$ | $^2/_{10}$ | $^1/_{10}$ | $\mathbf{^7/_{10}}$ |
| Rearrange | $^2/_{10}$ | $^3/_{10}$ | $^6/_{10}$ | $^4/_{10}$ | $\mathbf{^7/_{10}}$ |
| All | $^{10}/_{30}$ | $^{10}/_{30}$ | $^{13}/_{30}$ | $^{11}/_{30}$ | $\mathbf{^{23}/_{30}}$ |

**Table 2: Real robot results.** Our method MCR performs best in all tasks. Each method is fairly assessed over 10 trials on each task.

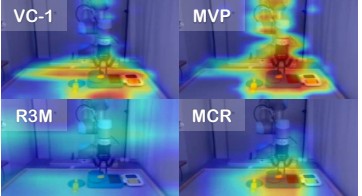

**Figure 8: Grad-CAM on Rearrange.** MCR is with best manipulation centricty.

We also visualize the Grad-CAM results for each method on the challenging Rearrange task in Figure 8. Representations such as R3M fail to focus on key manipulation regions, resulting in poor performance. MVP and VC-1 perform better, as they can capture the robot arm and relevant objects on the table, but they tend to over-focus on irrelevant items such as the pot. Our method avoids these pitfalls, achieving the best performance across all tasks. In conclusion, these real-world experiments underscore the effectiveness of our method in enhancing robot manipulation capabilities, significantly outperforming baseline approaches in challenging scenarios.

### 5.3 ABLATION STUDIES

The results presented in Table 4 indicate the effect of key design choices in our approach, offering insights and guidance when employing our manipulation-centric representations. All experiments are conducted on three challenging tasks: Can from Robomimic, Stick Pull from MetaWorld, and Laptop from DexArt, covering all robot arms and end-effectors in prior simulation experiments.

**All objectives improve manipulation centricity.** We begin by evaluating the effect of each training objective outlined in Equation (4) through an ablation study. Our results reveal that all objectives are essential to achieving strong downstream performance. In particular, the dynamics alignment and action prediction losses have the most significant effect. We attribute this to their ability to effectively utilize dynamics-relevant information in the robot dataset, which enhances manipulation centricity. Additionally, the temporal contrastive loss also plays a crucial role by capturing the temporal dependencies in the video sequences. Collectively, this analysis supports our key claim: learning manipulation-centric representations is beneficial for robotic manipulation, and our proposed objectives substantially improve this capability.

**Table 4: Key design choices of MCR.**

| Ablated Components | Success Rate (%) |
|--------------------|------------------|
| **Training Objective** | |
| w/o. objective $\mathcal{L}_{\text{dyn}}$ | 66.2 ($\pm 0.8$) |
| w/o. objective $\mathcal{L}_{\text{act}}$ | 71.3 ($\pm 1.2$) |
| w/o. objective $\mathcal{L}_{\text{tcl}}$ | 72.0 ($\pm 1.2$) |
| **Dynamic Chunk** | |
| Length $l$: 3→1 | 72.1 ($\pm 2.9$) |
| Length $l$: 3→5 | 76.8 ($\pm 2.4$) |
| Length $l$: 3→7 | 76.8 ($\pm 2.2$) |
| **Encoder Backbone** | |
| ResNet-: 50→18 | 77.3 ($\pm 1.8$) |
| ResNet-: 50→34 | 77.9 ($\pm 1.7$) |
| MCR (original) | **83.2** ($\pm 1.3$) |

**Medium dynamic chunk length works best.** Recall that dynamic chunks define the temporal horizon of robot state-action pairs utilized for modeling robot dynamics, as specified in Equation (1).

Our experiments indicate that a medium chunk length of 3 yields optimal performance. In contrast, a shorter chunk length (i.e., $l = 1$) fails to adequately capture action information, resulting in an insufficient representation of the underlying dynamics. Additionally, the size of a single-state chunk is considerably smaller than that of the pre-trained visual features, which can introduce noise when projected through a multi-layer perceptron (MLP). On the other hand, excessively long chunks may complicate the accurate modeling of dynamics. This study highlights the necessity of effectively encoding dynamic information in representations to enhance manipulation centricity.

**Larger encoders lead to better performance.** Finally, we also observe a clear trend: larger encoders consistently result in better performance. We hypothesize that larger models have a greater capacity to comprehend complex scenes and capture critical information related to dynamics and manipulation centricity. This finding is consistent with the ones in previous work (Shang et al., 2024), suggesting that our method scales well with the model size and has the potential for even greater improvements in performance if larger models are employed.

## 5.4 Analysis on Robot Dataset

Besides the studies on methodology design, we also reveal more insights into the utilization of robot dataset in learning our robotic representations.

**Larger dataset, better performance.** We begin by examining how the size of the robot dataset affects pre-training outcomes. Specifically, we reduced the data in each DROID scene from 100% to 25% and assessed the downstream performance, as shown in Figure 9. Our findings indicate that larger datasets contribute to improved representation quality. This seemingly contrasts with previous research (Dasari et al., 2023), which suggested that merely increasing dataset size may not yield benefits. We remark that the key differences here are (1) our use of robot data rather than human data, and (2) the incorporation of additional robot dynamics information. These factors enhance the scalability and effectiveness of MCR when applied to larger robot datasets. While some studies have attempted to combine human and robot data (Dasari et al., 2023; Majumdar et al., 2023), little improvement was observed, likely due to the insufficient utilization of robot dynamics. In summary, our method effectively scales with dataset size by leveraging dynamics information.

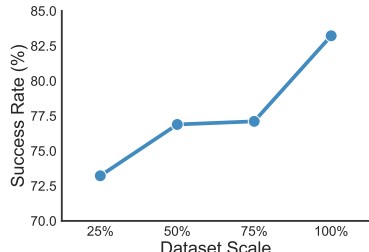

**Figure 9: Effect of robot dataset size.**

**Greater benefits for tasks with less embodiment gap.** Next, we investigate which downstream tasks benefit more from pre-training with robot data. Specifically, we examine the embodiment gap associated with the end-effector. The simulation tasks are accordingly categorized into gripper-based and dexterous hand-based tasks. As illustrated in Figure 10, both R3M-DROID and MCR outperform R3M on gripper-based tasks in terms of manipulation centricity and downstream success rate, supporting our earlier conclusions. However, performance drops in hand-based tasks, where R3M-DROID even underperforms R3M. This is likely due to the fact that all data in DROID was collected using a gripper, which presents an embodiment gap compared to dexterous hands. To alleviate this issue, we suggest two attempts in the future: (1) leveraging dynamics from robot data better to mitigate the embodiment gap, and (2) incorporating more end-effectors, such as dexterous hands, into robot datasets to enhance our manipulation-centric representations.

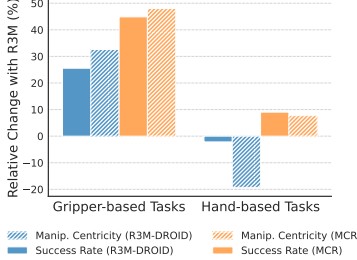

**Figure 10: Downstream domain gap.**

**Feature analysis.** To gain a more intuitive understanding of the impact of the robot dataset, we employ t-SNE (Van der Maaten & Hinton, 2008) to process and visualize feature embeddings generated by the pre-trained encoder. The visualization results are presented in Figure 11. In the simulation domain, R3M struggles to cluster images within individual tasks. However, this issue is partially alleviated by using a robot dataset, as R3M-DROID exhibits improved clustering ability. Nonetheless, due to the significant domain gap between DROID and simulation environments, it still encounters difficulties in distinguishing many tasks. In contrast, our method demonstrates markedly superior clustering capabilities, indicating the importance of incorporating real-robot dynamics for effective robotic representation. Additionally, we observe that all three methods exhibit good clustering performance in real-world tasks, likely attributed to the more distinct visual changes between tasks

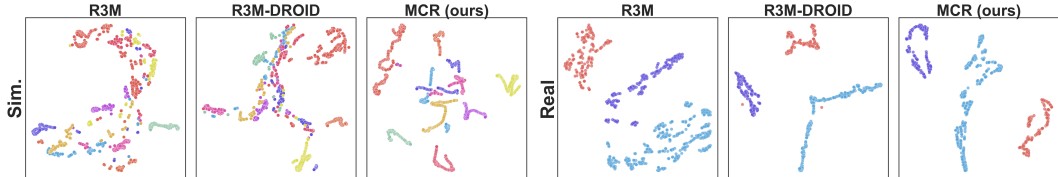

**Figure 11: t-SNE visualization.** We do t-SNE visualization on 10 simulation tasks from MetaWorld and 3 real robot tasks. Each dot represents an image frame and each color indicates a task. The results demonstrate that (1) our representation has the best clustering ability and (2) robot data is helpful to robotic representation.

compared to simulations. However, R3M remains inferior to the other two methods, reinforcing the critical role of robot datasets in enhancing robotic representations.

# 6 RELATED WORKS

**Pretrained robotic representations.** The development of pre-trained visual representations (PVRs) has significantly improved the efficiency of downstream policy learning in robotics. Notable approaches include a variant of MoCo-v2 (Parisi et al., 2022) that integrates multi-layer features, MVP (Xiao et al., 2022) and VC-1 (Majumdar et al., 2023) utilizing Masked Autoencoders (He et al., 2022), and R3M (Nair et al., 2022), which employs a time-contrastive objective and video-language alignment. Other works like Zheng et al. (2024) use temporal action contrastive learning, while MPI (Jia et al., 2024) focuses on predicting transition frames based on language instructions. HRP (Srirama et al., 2024) extracts affordances from large-scale human videos for enhanced generalization, and Theia (Shang et al., 2024) distills diverse vision models for robot learning. VIP (Ma et al., 2023) generates dense reward functions for robotic tasks. Similar to our method, RPT (Radosavovic et al., 2023) employs trajectory labels for representation training. In contrast, our work introduces the concept of manipulation centricity, leveraging large-scale robotic data to capture manipulation-specific dynamics, resulting in improved performance on downstream tasks.

**Learning from large-scale robotic data.** Recent advancements in robotics increasingly utilize large-scale datasets to enhance the capabilities of robotic systems. Collaboration et al. (2024) introduces the Open X-Embodiment dataset, the largest robotic dataset comprising extensive data from diverse robot embodiments across various tasks. The RT-X model (Collaboration et al., 2024), trained on this diverse dataset, shows promising results in cross-robot skill transfer as a generalist. Team et al. (2024) introduces Octo, a transformer-based diffusion policy also trained on Open X-Embodiment, supporting flexible task and observation definitions. Additionally, OpenVLA (Kim et al., 2024) is developed as a vision-language model enabling direct mapping from image inputs to continuous robot actions. Unlike previous approaches (Liang et al., 2024; Zheng et al., 2025), our work focuses on extracting dynamics and interaction information from robotic datasets to create specialized visual representations for policy learning, offering an efficient alternative to generalist policies. Other related works about dynamics-aware representations and factors driving effectiveness in robotic representations are presented in Appendix A.

# 7 CONCLUSIONS AND DISCUSSIONS

Our work introduces the concept of manipulation centricity in visual representations for robotic manipulation, revealing its crucial role in downstream task performance. By leveraging large-scale robot data and dynamics-aware learning objectives, we develop a method that significantly enhances the extraction of manipulation-centric features. This approach not only advances the state-of-the-art in robotic manipulation across diverse tasks but also provides a new lens through which to understand and evaluate representation learning in robotics. Our findings highlight the importance of aligning representation learning with the specific demands of robotic control, potentially shifting the paradigm of how we approach feature extraction for embodied agents. Looking forward, this work opens avenues for exploring multi-modal integration, such as using language instructions to learn task-aware features and further leveraging trajectory data to capture spatial-temporal robotic dynamics, promising to further narrow the gap between pre-trained generalized representation models and real-world robotic manipulation.

ACKNOWLEDGEMENTS

This project was supported by National Key R&D Program of China (2022ZD0161700). We would like to thank Zhecheng Yuan, Tianming Wei, Yanjie Ze, and Yuanhang Zhang for their insightful discussions and technical supports.

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

## A    ADDITIONAL RELATED WORKS

**Dynamics-aware representation.**  Representation learning is crucial for extracting key features from image inputs in imitation learning and reinforcement learning (Xu et al., 2023; Ji et al., 2024; Yuan et al., 2022). CURL (Laskin et al., 2020) uses InfoNCE (van den Oord et al., 2019) to maximize agreement between augmented observations, while CPC (Henaff, 2020) and ATC (Stooke et al., 2021) incorporate temporal dynamics into contrastive loss. DRIML (Mazoure et al., 2020) proposes a policy-dependent auxiliary objective, and KOROL (Chen et al., 2024) trains feature extractors on task-specific RGBD images during Koopman operator dynamics learning. TACO (Zheng et al., 2023; 2024) optimizes mutual information between current state-action pairs and future states. In contrast, our approach trains a generalized robotic representation model to extract more effective dynamics-aware representations, offering improved efficiency and generalization through a train-once-for-all methodology.

**Factors driving effectiveness in robotic representation.**  Burns et al. (2024) identifies emergent segmentation ability as a critical factor in the success of pre-trained visual models for robotic manipulation. While their focus is on generalization capability, our emphasis is on the downstream learning efficiency of robotic representations, investigating this from the perspective of manipulation centricity. Additionally, Theia (Shang et al., 2024) highlights the importance of high entropy in feature norm distribution for performance enhancement. Although they validate their findings in simulation locomotion tasks, we evaluate the effectiveness of manipulation centricity in both simulated and real-world robot manipulation tasks.

## B    TRAINING EFFICIENCY

Pre-training representations typically requires extensive computational resources. Methods like VC-1 and MVP have prolonged training times due to their use of MAE (Feichtenhofer et al., 2022). Even the most computationally efficient baseline, R3M, requires 120 hours on a single NVIDIA V100. Our method, however, achieves state-of-the-art performance while requiring less computation. As shown in Table 5, our approach has a shorter training time than R3M, achieving an good balance between performance and computational efficiency.

**Table 5: Computation efficiency.** Training computation requirements across methods.

| Method | GPU Type | Training Time (h) |
|--------|----------|-------------------|
| R3M | Tesla V100 | $\approx 120$ |
| MCR | RTX 3090 Ti | $\approx 50$ |

## C    MORE EXPERIMENTAL DETAILS

### C.1    TASK DESCRIPTIONS

We select a diverse set of tasks from various robotic manipulation benchmarks for evaluation. Specifically, we include three tasks from Robomimic (Mandlekar et al., 2021), three tasks from RoboCasa (Nasiriany et al., 2024), ten tasks from MetaWorld (Yu et al., 2019), and four dexterous tasks from DexArt (Bao et al., 2023). Detailed descriptions of each task are provided below:

- Can (Robomimic, $\mathcal{A} \in \mathbb{R}^7$): the task is to manipulate the can using the robot's arm to perform various actions such as picking it up, moving it to a different location, and orienting it in a specific way.
- Lift (Robomimic, $\mathcal{A} \in \mathbb{R}^7$): the task is to grasp a specified item and then raise it to a desired height.
- Square (Robomimic, $\mathcal{A} \in \mathbb{R}^7$): the task is to pick up a square-shaped nut and place it onto a rod successfully.
- Close Drawer (RoboCasa, $\mathcal{A} \in \mathbb{R}^7$): the task is to accurately close a drawer.
- Coffee Button Press (RoboCasa, $\mathcal{A} \in \mathbb{R}^7$): the task is to press the button on the coffee machine to start the coffee brewing process.
- Open Single Door (RoboCasa, $\mathcal{A} \in \mathbb{R}^7$): the task is to open a door that is singularly paneled, such as a cabinet or microwave door, which is already closed.
- Assembly (MetaWorld, $\mathcal{A} \in \mathbb{R}^4$): the task is to grasp a nut and position it on a dowel using the gripper.
- Bin Picking (MetaWorld, $\mathcal{A} \in \mathbb{R}^4$): the task is to transfer a disc from one bin container to another.

- `Button Press` (MetaWorld, $\mathcal{A} \in \mathbb{R}^4$): the task is to press a button using a robotic arm to activate a device.
- `Disassemble` (MetaWorld, $\mathcal{A} \in \mathbb{R}^4$): the task is to remove a nut from a peg by picking it up.
- `Drawer Open` (MetaWorld, $\mathcal{A} \in \mathbb{R}^4$): the task is to accurately open a drawer.
- `Hammer` (MetaWorld, $\mathcal{A} \in \mathbb{R}^4$): the task is to drive a screw into the wall using a hammer.
- `Pick Place Wall` (MetaWorld, $\mathcal{A} \in \mathbb{R}^4$): the task is to grab a puck, go around a wall and put the puck in the designated spot.
- `Shelf Place` (MetaWorld, $\mathcal{A} \in \mathbb{R}^4$): the task is to grab a puck and set it on a shelf.
- `Stick Pull` (MetaWorld, $\mathcal{A} \in \mathbb{R}^4$): the task is to use a stick to pull a box by holding onto the stick.
- `Stick Push` (MetaWorld, $\mathcal{A} \in \mathbb{R}^4$): the task is to hold a stick to push a box with it.
- `Bucket` (DexArt, $\mathcal{A} \in \mathbb{R}^{22}$): the task is to elevate a bucket..
- `Faucet` (DexArt, $\mathcal{A} \in \mathbb{R}^{22}$): the task is to activate a faucet using a rotating joint.
- `Laptop` (DexArt, $\mathcal{A} \in \mathbb{R}^{22}$): the task is to grasp the center of the display and then lift the laptop cover.
- `Toilet` (DexArt, $\mathcal{A} \in \mathbb{R}^{22}$): the task is to initiate the process of lifting a bigger toilet seat.

## C.2 MORE GRAD-CAM VISUALIZATIONS

The Grad-CAM visualizations for each task are presented in Table 8, which provides a comprehensive comparison with other baseline methods. Notably, our approach is shown to effectively facilitate the capture of key manipulation-centric features, thereby enhancing the model's ability to focus on the most relevant aspects of the task.

## C.3 PRE-TRAINING HYPERPARAMETERS

We show our hyperparameters during the pre-training stage in Table 6. Downstream policy learning settings are introduced in Section C.5.

**Table 6:** Hyperparameters for MCR pre-training.

| Hyperparameter | Value |
|---|---|
| Encoder type | ResNet50 |
| Batch size | 32 |
| Learning rate | 1e-4 |
| Training steps | 500,000 |
| Data augmentation | RandomResizedCrop (224,(0.5, 1.0)) |
| Optimizer | Adam |
| DROID views used | two exterior views |
| DROID proprioception used | cartesian and gripper position |

## C.4 PRE-TRAINING IMPLEMENTATION

Our codebase is built upon the implementation of R3M. Similarly, for one sample within the batch, we have 5 frames from one video. The initial and final frames are sampled from the first and last 20% of the video, respectively. We employ the same contrastive learning loss implementation, modifying only the positive and negative sample pairs. For more details, please refer to Nair et al. (2022).

To clarify further, we provide our PyTorch-like code implementation of $L_a$ as below:

```
actor_trunk =
    # self.outdim is the output dimension of ResNet, for example, ResNet50
        ↪ is 2048
    nn.Sequential(nn.Linear(self.outdim, 50),
    nn.LayerNorm(50), nn.Tanh())

actor_policy = nn.Sequential(
    # action_dim in our case in 7 for DROID dataset
```

```
nn.Linear(50, 512),
nn.ReLU(inplace=True),
nn.Linear(512, 512),
nn.ReLU(inplace=True),
nn.Linear(512, action_dim))
```

The MLP part of dynamics alignment loss $L_d$ is implemented as follows:

```
# calculate the length of state-action dynamics chunk
state_input_dim = 14 * self.state_chunk_length # state
state_input_dim += 7 * (self.state_chunk_length - 1) # action

state_encoder = nn.Sequential(
    nn.Linear(state_input_dim, 1024),
    nn.ReLU(),
    nn.Linear(1024, self.outdim))
```

### C.5   BC Implementation and Settings

**MetaWorld and DexArt.** We evaluate our model on these two domains following (Huang et al., 2024) and (Ze et al., 2024). Scripted policies are used to generate demonstrations in MetaWorld, and policies trained with Reinforcement Learning is used in DexArt. We generate 25 demonstrations per task in MetaWorld and 100 in DexArt, where robot end effectors and objects are initially randomized. The downstream BC policy is a three-layer MLP with ReLu activations and hidden sizes of 256. A BatchNorm layer is added before the feature is input into the MLP. The policy is trained with a mean squared error (MSE) loss, a learning rate of 0.001, and a batch size of 256.

**Robomimic.** We adopt the released behavior cloning implementation from RoboMimic, and use their standard imitation learning dataset. The dataset contains 200 demonstrations for each task. We only modify the code to evaluate diverse visual encoders. The downstream policy is a two-layer MLP with hidden sizes of 1024. The policy is trained with an MSE loss, an initial learning rate of 0.001, and a batch size of 100. The learning rate is decayed with a factor of 1.

**RoboCasa.** We utilize the official policy learning implementation from RoboCasa. We use the BC-Transformer algorithm implemented by RoboMimic, with a RoboCasa-standard configuration as reported in the appendix of paper Nasiriany et al. (2024). We modify the visual observation encoder and train BC policies using the "Human-50" dataset collected by human operators. The policy is trained with an MSE loss, an initial learning rate of 0.0001, and a batch size of 16. The learning rate is decayed with factor 1.

**Real world** Our real-world codebase is adapted from V-D4RL. We collect demonstrations using a keyboard interface, with 40 demonstrations for Sweep and Rearrange, and 30 for Lift due to its simplicity. The policy is a 3-layer BatchNormMLP with a hidden size of 1024. We train the policy using the log-likelihood loss, with a learning rate of 0.0001 and a batch size of 256.

### C.6   Behavior Cloning (BC) Performance of Each Simulation Task

We show the performance per task of our main results in table 7. Our method achieves the highest success rate in 19 out of 20 tasks evaluated, showcasing its robustness and adaptability to various scenarios.

## D   More details on Manipulation Centricity

**Grad-CAM.** Grad-CAM is a widely used technique for generating heatmaps of input images to identify the regions that the encoder focuses on. We utilize the PyTorch-Grad-CAM library to generate Grad-CAM figures for each visual encoder. In convolutional neural networks (CNNs), Grad-CAM generates heatmaps by backpropagating gradients through the final convolutional layers, thereby highlighting important image regions. However, in Vision Transformers (ViT), the final classification is based on the class token computed in the last attention block, which means that the output is not influenced by the 14×14 spatial channels in the final layer, resulting in zero gradients. To generate meaningful visualizations in ViT, it is necessary to choose a layer before the final attention block. In our study, we chose the LayerNorm applied to the output of the self-attention mechanism in the last Transformer block.

**Table 7: Main results on 20 simulation tasks.** Results for each task are provided in this table. A summary across domains is shown in Figure 6.

| Alg \ Task | DexArt | | | | Robomimic | | | RobocCasa | | |
|---|---|---|---|---|---|---|---|---|---|---|
| | Bucket | Faucet | Laptop | Toilet | Can | Lift | Square | CloseDrawer | CoffeeButtonPress | OpenSingleDoor |
| MCR (ours) | **36.7** (±2.9) | **38.3** (±2.9) | **93.3** (±2.9) | 73.3 (±2.9) | **68.0** (±4.0) | **96.0** (±2.3) | **30.0** (±1.2) | **99.3** (±1.2) | **72.0** (±3.5) | **56.0** (±3.5) |
| LfS | 33.3 (±5.8) | 36.7 (±5.8) | 83.3 (±10.4) | 71.7 (±2.9) | 6.0 (±0.0) | 64.0 (±4.2) | 4.0 (±0.0) | 85.3 (±1.2) | 52.0 (±4.0) | 46.7 (±1.2) |
| MVP | 31.7 (±2.9) | 33.3 (±2.9) | 81.7 (±5.8) | **80.0** (±0.0) | 28.0 (±2.0) | 74.0 (±6.4) | 14.0 (±2.3) | 98.0 (±2.0) | 52.7 (±18.9) | 33.3 (±14.5) |
| VC1 | 30.0 (±0.0) | 35.0 (±0.0) | 85.0 (±0.0) | 71.7 (±2.9) | 44.0 (±7.0) | 74.0 (±9.2) | 20.0 (±3.5) | 98.7 (±2.0) | 29.3 (±5.8) | 33.3 (±7.0) |
| R3M | 31.7 (±2.9) | 36.7 (±2.9) | 81.7 (±5.8) | 71.7 (±2.9) | 50.0 (±4.2) | 86.0 (±6.0) | 24.0 (±1.2) | 88.7 (±3.1) | 47.3 (±6.1) | 48.7 (±7.6) |
| HRP | 31.7 (±2.9) | 36.7 (±2.9) | 90.0 (±5.0) | 63.3 (±14.4) | 42.0 (±3.5) | 86.0 (±3.5) | 26.0 (±2.3) | 91.3 (±4.6) | 35.3 (±11.6) | 38.0 (±6.0) |
| R3M-Droid | 35.0 (±5.0) | 33.3 (±2.9) | 80.0 (±0.0) | 66.7 (±7.6) | 54.0 (±2.3) | **96.0** (±0.0) | 22.0 (±3.1) | 88.7 (±2.3) | 51.3 (±2.3) | 45.3 (±7.6) |

| Alg \ Task | MetaWorld | | MetaWorld (Medium) | | MetaWorld (Hard) | MetaWorld (Very Hard) | | | | |
|---|---|---|---|---|---|---|---|---|---|---|
| | Button Press | Drawer Open | Bin Picking | Hammer | Assembly | Shelf Place | Disassemble | Stick Pull | Stick Push | Pick Place Wall |
| MCR (ours) | **100.0** (±0.0) | **100.0** (±0.0) | **96.7** (±2.9) | **100.0** (±0.0) | **100.0** (±0.0) | **41.7** (±5.8) | **93.3** (±2.9) | **86.7** (±2.9) | **100.0** (±0.0) | **91.7** (±2.9) |
| LfS | 96.7 (±2.9) | 95.0 (±5.0) | 81.7 (±2.9) | 95.0 (±5.0) | 95.0 (±5.0) | 35.0 (±5.0) | 86.7 (±2.9) | 83.3 (±5.8) | 96.7 (±2.9) | 85.0 (±5.0) |
| MVP | 96.7 (±2.9) | 98.3 (±2.9) | 81.7 (±2.9) | 91.7 (±2.9) | 86.7 (±2.9) | 20.0 (±5.0) | 65.0 (±8.7) | 75.0 (±8.7) | 96.7 (±2.9) | 76.7 (±11.6) |
| VC-1 | 98.3 (±2.9) | 98.3 (±2.9) | 78.3 (±2.9) | 86.7 (±2.9) | 95.0 (±5.0) | 21.7 (±2.9) | 66.7 (±2.9) | 86.7 (±2.9) | 98.3 (±2.9) | 71.7 (±2.9) |
| R3M | 91.7 (±2.9) | 71.7 (±16.1) | 21.7 (±2.9) | 63.3 (±5.8) | 36.7 (±2.9) | 35.0 (±8.7) | 76.7 (±2.9) | 43.3 (±7.6) | 71.7 (±2.9) | 58.3 (±5.8) |
| HRP | 98.3 (±2.9) | 98.3 (±2.9) | 90.0 (±0.0) | 65.0 (±0.0) | 96.7 (±2.9) | 23.3 (±2.9) | 61.7 (±2.9) | 85.0 (±0.0) | 96.7 (±2.9) | 81.7 (±2.9) |
| R3M-Droid | 98.3 (±2.9) | 96.7 (±5.8) | 90.0 (±0.0) | 80.0 (±0.0) | 83.3 (±5.8) | 38.3 (±2.9) | 66.7 (±2.9) | 61.7 (±20.2) | 98.3 (±2.9) | 83.3 (±5.8) |

Our paper applies Grayscale CAM to measure Manipulation Centricity, where the pixel values range from 0 (black) to 255 (white), with higher values (closer to white) indicating regions that contribute more significantly to the model's decision. Furthermore, we binarize the Grayscale CAM by thresholding pixel values at 2, setting values below 2 to 0 and those above or equal to 2 to non-zero values, thereby mitigating the effect of noise and facilitating more accurate interpretations and precise metric calculations.

**SAM 2.** Segment Anything Model 2 (SAM 2, Ravi et al. (2024)) is a large-scale segmentation model. In our study, we utilize SAM2 to segment demonstration videos of all simulation tasks. Specifically, we select the official 'base plus' model available on the SAM2 repository and adopt the implementation of the SAM2-GUI, an interactive graphical user interface to utilize pre-trained SAM2 models. We manually upload each task video and add prompt points to generate segmentation videos. Notably, foreground pixels, including robot end effectors and objects, are labeled with non-zero values, while background pixels are labeled with zero. A full list of ground truth annotations is shown in Table 8, to better illustrate manipulation centricity.

**Jaccard Index.** We employ the Jaccard Index to quantify the similarity between Grad-CAM and SAM2 video frames, with the index ranging from 0 to 1. A higher Jaccard Index value indicates a greater degree of similarity between the two, suggesting a stronger alignment between the attention maps generated by Grad-CAM and the segmentation masks produced by SAM2.

# E   DETAILED ANALYSIS OF DROID SUBSET USED

As described in Section 4.1, we perform a preliminary preprocessing of the full 1.7TB RLDS dataset, yielding a subset that serves as the basis for our analysis. In this section, we present a statistical characterization of this subset. Specifically, we exclude videos with fewer than 40 frames. The distribution of video lengths in the subset is illustrated in Figure 12a. Furthermore, we conduct a comprehensive statistical analysis of the language instructions accompanying the dataset, with a particular focus on the frequency of common object nouns and action verbs. The results of this analysis are visualized in Figures 12b and 12c, respectively. Notably, our subset covers a large degree of data diversity, capturing a wide range of scenarios, objects, and actions, thereby ensuring the effectiveness of our method.

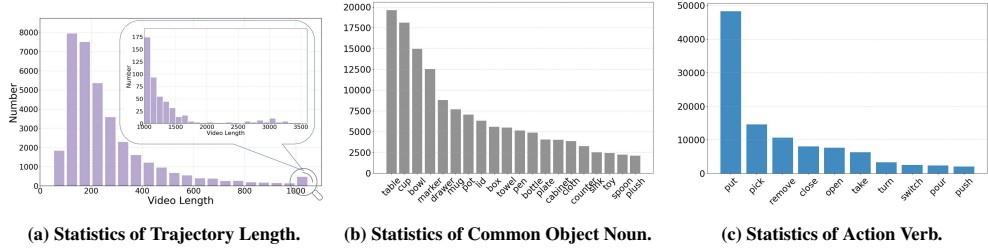

(a) Statistics of Trajectory Length.    (b) Statistics of Common Object Noun.    (c) Statistics of Action Verb.

Figure 12: Statistical Analysis of the DROID Subset.

Table 8: Grad-CAM of all tasks.

Continue

**Table 8: Grad-CAM of all tasks**

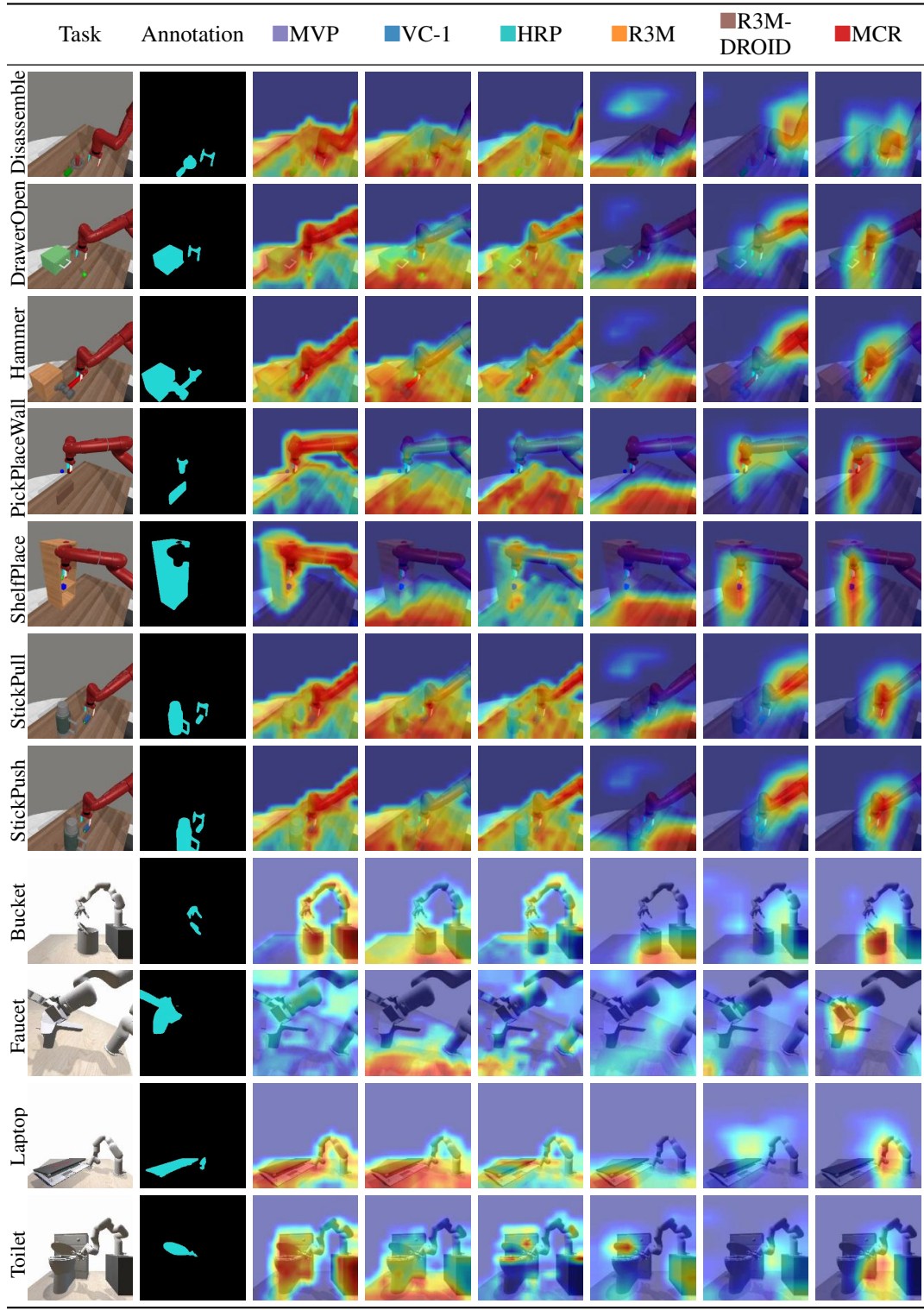

