# OpenReview forum: "Robots Pre-train Robots: Manipulation-Centric Robotic Representation from Large-Scale Robot Datasets"
_ICLR.cc/2025/Conference — ICLR 2025 Poster_

### Official Review · Reviewer_vJK6 · 2024-10-21

**Soundness:** 2
**Presentation:** 3
**Contribution:** 2
**Rating:** 3
**Confidence:** 4

**Summary:**

This paper presents a representation learning approach named RPM that learns to encode images into low-dimensional states that are aligned with the proprioception state through contrastive learning.

1. The authors showed that RPM outperforms existing learned robot representations on 20 tasks selected from 4 benchmarks.
2. The authors proposed to use gradient heatmap as a heuristic named as "manipulation-centricity" to determine how much the learned representation contributes to the final manipulation task performance.
3. RPM is showed to have highest manipulation-centricity than existing learned robot representations, and is evaluated with a real UR5 robot in lift, sweep and rearrange tasks.

**Strengths:**

1. The proposed heuristics of using gradient heatmap to indicate manipulation performance is interesting. It can be linked to existing work in visual affordance prediction.
2. Aligning the vision representation with the robot proprio state and actions is interesting, and may encourage the model to focus more on task-relevant information.
3. The writing is easy to understand.

**Weaknesses:**

1. The technical contribution is limited. The authors spent 4 pages between the introduction and experiment section to discuss their method. But, in fact, the only new technical point that the authors proposed is equation (1), which uses InfoNCE loss to bring visual features and proprio state/aciton closer. This looks hand-wavy. Even this innovation raises questions, as the proprio state/action only describe the robot trajectory but not the visual scenes. This is very likely to make the model overfit to the learned scenarios.

2. The evaluation is questionable. RPM is evaluated on 20 tasks, 10 from MetaWorld, 4 from DexArt, 3 from RoboCasa, 3 from RoboMimic. However, these benchmarks combined include hundreds of tasks. Why just select these 20 particular tasks? Moreover, the "Can" task in robomimic is mentioned as challenging in this paper, but it is just a simple pick-and-place task of a moderate-size can. The evaluation raises serious questions about the generalization of this RPM method.

2. Unnecessary discussion on how to use the human video datasets. The authors spend a considerable amount of efforts to explain the policy learning from human videos. But in the end, the conclusion is to not use human data. Human action data is abundant and contains rich and diverse behavior. If the authors' proposal is to not use them and use robot demonstrations, which in fact is the standard robot learning setup. I would prefer the paper to focus on the studied setting.

3. Needs a more comprehensive ablation study. RPM is pre-trained with the DROID dataset, a large and relatively clean robot dataset, while many baselines that RPM compared against are not trained with DROID.

4. In Table.1, HRP and VC-1 achieves better success rate than R3M-DROID, yet R3M-DROID's gradient heatmap focus more on the objects. According to the key finding, R3M-DROID shall perform better.

4. The term "dynamics" is mis- and over-used, confusing in some sections.

**Questions:**

(see weakness)

---

> ### Author Response · Authors · 2024-11-22
> **Rebuttal to Reviewer vJK6**
>
> We sincerely appreciate Reviewer vJK6's positive feedback regarding both our paper's presentation and our conceptual contributions. Below, we provide detailed responses to address the reviewer's concerns:
>
> > w1: The technical contribution is limited. The authors spent 4 pages between the introduction and experiment section to discuss their method. But, in fact, the only new technical point that the authors proposed is equation (1), which uses InfoNCE loss to bring visual features and proprio state/aciton closer. This looks hand-wavy.
>
> We respectfully disagree with the assessment of limited technical contribution, and would like to highlight two key technical aspects of our work:
>
> * Following the introduction, we present a detailed analysis of how the manipulation-centricity of pretrained encoders impacts downstream task performance, introducing a clear and practical metric. This novel contribution has been specifically recognized by **Reviewers wcpD, 3aRj, and cSN9**. We emphasize that we are **the first to systematically examine and quantify this critical problem in robotic manipulation.**
>
> * Beyond the technical contribution of state-action alignment mentioned by the reviewer, in our knowledge, we are **the first to incorporate action prediction loss in visual encoder pretraining for manipulation tasks**. We appreciate any relevant prior work Reviewer vJK6 may suggest that we have missed, as this would help us better contextualize our contribution and strengthen our related work discussion.
>
> Thank you for your feedback. We believe these technical contributions constitute significant components of our work and hope this clarification addresses your concerns.
>
> > w1: Even this innovation raises questions, as the proprio state/action only describe the robot trajectory but not the visual scenes. This is very likely to make the model overfit to the learned scenarios.
>
> Our approach intentionally leverages robot trajectory information to capture effective visual features. As emphasized in our paper, our training utilizes a highly diverse dataset comprising **36,000 trajectories across more than 500 distinct scenes and 80 different tasks**. This extensive data diversity substantially reduces the risk of overfitting. Moreover, **our empirical evaluation demonstrates robust generalization across various robot embodiments (beyond the Franka robot used in the DROID dataset)** and diverse manipulation tasks in both simulations and real world, providing strong evidence that our pretrained encoder successfully avoids overfitting issues.
>
> > w2: The evaluation is questionable. RPM is evaluated on 20 tasks, 10 from MetaWorld, 4 from DexArt, 3 from RoboCasa, 3 from RoboMimic. However, these benchmarks combined include hundreds of tasks. Why just select these 20 particular tasks? Moreover, the "Can" task in robomimic is mentioned as challenging in this paper, but it is just a simple pick-and-place task of a moderate-size can.
>
> We would like to clarify our task selection and address the concerns about task difficulty:
>
> * For **MetaWorld**, we selected the five most difficult tasks according to the benchmark's difficulty ratings, plus five tasks evaluated in R3M.
>
> * For **DexArt**, we evaluated all four available tasks in the benchmark.
>
> * The limited evaluation on **RoboCasa** (3 tasks) was due to practical constraints: very slow inference speed in their environment and incomplete demonstration data.
>
> * For **RoboMimic**, most tasks are challenging, with near-zero performance across existing algorithms. We selected the most commonly evaluated tasks in the literature.
>
> In summary, **our evaluation suite is more comprehensive than all baselines, spanning multiple major manipulation benchmarks and real-world tasks.** ​​While the "Can" task may appear simple as a pick-and-place operation, its difficulty is evidenced by the poor performance of most methods. As widely discussed in manipulation works [1][2], pick-and-place tasks are far from trivial in robotics, because task difficulty isn't solely determined by motion complexity but also by environmental factors and perception challenges.
>
> [1] Kim et al. "OpenVLA: An Open-Source Vision-Language-Action Model"
>
> [2] Octo Model Team et al. "Octo: An Open-Source Generalist Robot Policy"

---

> ### Author Response · Authors · 2024-11-22
> **Rebuttal to Reviewer vJK6**
>
> > w3: Unnecessary discussion on how to use the human video datasets. The authors spend a considerable amount of efforts to explain the policy learning from human videos. But in the end, the conclusion is to not use human data. Human action data is abundant and contains rich and diverse behavior. If the authors' proposal is to not use them and use robot demonstrations, which in fact is the standard robot learning setup. I would prefer the paper to focus on the studied setting.
>
> We would like to clarify that our contribution extends beyond simply training a pretrained model on robotic datasets, which would indeed only require a discussion of robot data. **One of the key contributions of our work is proposing a manipulation-centric metric for evaluating all visual encoders**, as other reviewers specifically recognize this contribution.  Through our discussion of models trained on human data, we analyze how baseline models insufficiently capture manipulation-related features. This analysis demonstrates that training on robot data may be more effective in enhancing manipulation centricity. **Such analysis is essential to establish our key insight and justify our approach to answer how to pretrain on robotic data more effectively for manipulation tasks**, rather than simply assuming this choice. We hope this clarification helps better convey our motivation and contribution.
>
> > w4: Needs a more comprehensive ablation study. RPM is pre-trained with the DROID dataset, a large and relatively clean robot dataset, while many baselines that RPM compared against are not trained with DROID.
>
> Thank Reviewer vJK6 for this important suggestion about ablation studies. To address this concern, while R3M's performance on DROID was already included in our paper, we have further expanded our comparisons by including results from MAE (used in MVP and VC-1) pretrained on DROID. Our experiments show that MAE achieves limited performance when trained on DROID due to reduced image quantity and lack of robotics-specific design. **This further highlights the importance of designing representation learning methods specifically for robotics datasets.**
>
> Regarding the HRP baseline, it requires hand pose and object labels for dexterous manipulation, which are not available in DROID, making direct training on DROID infeasible. We have included these additional discussions and detailed results in Appendix A.7 for a more comprehensive comparison.
>
> |  task  |  LIV   | SAM2  |  MAE-DROID  | Ours  |
> |  ----  |  ----  | ----  | ----  | ----  |
> | can  |  4  |  5.3  |  4  |   68.3  |
> |  lift  |  86.7  |  60.7  |  50.7  |   98.7  |
> |  square  | 15.3  |  2  |  6.7  |  31.3  |
> |  aggregate  | 35.3  |  22.7  |  20.5  |  66.1  |

---

> > ### Author Response · Authors · 2024-11-22
> > **Rebuttal to Reviewer vJK6**
> >
> > > w5: In Table.1, HRP and VC-1 achieves better success rate than R3M-DROID, yet R3M-DROID's gradient heatmap focus more on the objects. According to the key finding, R3M-DROID shall perform better.
> >
> > We sincerely thank the reviewer for proposing this observation. Since the RGB-version Grad-CAM figures may have minor visual differences that are hard to distinguish for human eyes, we calculate the numerical results for the tasks in Table 1 as included in **Appendix A.12**. The results demonstrate strong correlation between manipulation centricity and success rates across tasks. In the Square task, most baselines (except MVP) show very similar manipulation centricity values corresponding to similar success rates. The Pick&Place task particularly validates our metric, showing clear correlation between manipulation centricity and performance. Table 1 also reveals that while R3M-DROID's representation focuses primarily on the end-effector, our method effectively captures key manipulation-related features including the end-effector and objects, demonstrating the effectiveness of our approach.
> >
> > |  task-metric  |  MVP   | VC-1  |  HRP  |  R3M  |  R3M-DROID  |  Ours  |
> > |  ----  |  ----  | ----  | ----  |  ----  | ----  | ----  |
> > |  square-manip. cen.  |  0.025  |  0.060  |  0.059  |  0.057  |  0.062  |  0.078  |
> > |  square-success rate  |  12  |  23  |  22  |  23  |  22  |  31  |
> > |  P&P Wall-manip. cen.  | 0.226  |  0.238  |  0.238  |  0.196  |  0.214  |  0.258  |
> > |  P&P Wal-success rate  | 86  |  96  |  95  |  36  |  83  |  100  |
> >
> > > The term "dynamics" is mis- and over-used, confusing in some sections.
> >
> > Thank you for this helpful comment on terminology precision. We have consistently defined dynamics labels in our paper as labels including actions and proprioception. While we used "dynamics" as a shorthand for "state-action dynamics labels," we agree that using the complete term "state-action dynamics" would provide better clarity. **We have updated the paper according to your suggestions.**
> >
> > We have addressed Reviewer vJK6's concerns point by point and enhanced our paper based on the valuable suggestions, particularly in clarifying our technical contributions and providing comprehensive ablation studies. We hope these responses have adequately addressed your questions and merit reconsideration of the initial score. Your feedback has been instrumental in improving our paper's quality, and we welcome any further discussion.

---

> > > ### Author Response · Authors · 2024-11-25
> > > **Does our response address your concerns?**
> > >
> > > Dear Reviewer vJK6,
> > >
> > > Thank you so much for your time and effort in reviewing our work and providing insightful feedback. As the review discussion phase concludes in two days, we have not yet received your feedback.
> > >
> > > Your feedback is crucial for us to ensure our response adequately addresses your concerns and improves the quality of our paper. Therefore, we sincerely request you to review our response and confirm if we have resolved your questions.
> > >
> > > We deeply appreciate your contribution to the community through your review. Your insights are invaluable to us. We eagerly await your response.
> > >
> > > Best Regards,
> > >
> > > Paper 793 Authors

---

> ### Author Response · Authors · 2024-11-28
> **Eagerly Awaiting Your Response, Reviewer vJK6**
>
> Dear Reviewer vJK6,
>
> As the discussion period draws to a close soon, we are reaching out to request your feedback on our response kindly.
>
> We are delighted to have received positive feedback from the other three reviewers and are very eager to ensure that our response has adequately addressed your concerns as well.
>
> We believe that the detailed clarification of our contributions, along with the additional experiments in rebuttal could solve your concerns. Your feedback is crucial for us to further enhance the quality of our paper. We eagerly await your response!
>
> Warm Regards,
>
> Paper 793 Authors

---

> > ### Author Response · Authors · 2024-12-01
> > **Awaiting Your Valuable Feedback (for the final 24 hours)**
> >
> > Dear Reviewer vJK6,
> >
> > Since today is the last day of the author-reviewer discussion period, we kindly request you to review our previous response and revised manuscript and let us know if there are any other questions.
> >
> > We are happy to answer any additional questions you may have. Thank you for your time and effort in reviewing our work.
> >
> > Best Regards,
> >
> > Paper 793 Authors

---

### Official Review · Reviewer_cSN9 · 2024-10-27

**Soundness:** 3
**Presentation:** 3
**Contribution:** 3
**Rating:** 8
**Confidence:** 4

**Summary:**

This paper introduces a metric (manipulation-centricity) for assessing the ability of pretrained visual encoders to model task-relevant features for robot manipulation tasks; as well as a novel pretraining objective (RPM) that incorporates the robot dynamics information available in robotics datasets. The authors first establish a correlation between the metric they define and the success rate of policies that leverage the features generated by pretrained visual encoders. Having established a positive correlation they seek to design a pretraining objective for visual encoders and evaluate its performance on their proposed metric and the downstream policy success rates. The novelty in the pretraining objective the authors introduce is a dynamics alignment term which takes the form of the InfoNCE applied to embeddings of a sequence of state action pairs and image embeddings. This novel dynamics alignment term is combined with existing behaviour cloning and temporal contrastive terms to give the full learning objective. They apply this objective function to pretrain visual encoders for manipulation tasks and demonstrate that this pretraining objective leads to improvements on their metric and downstream task performance in both simulated and real-world experiments.

**Strengths:**

- The paper validates a pretraining objective for visual encoders that improves the performance of robot policy learning.

- The paper introduces a novel metric for quantitatively analysing the features generated by visual encoders.

- The paper validates its claims in both simulated and real-world experiments.

- The paper includes analysis of the representations learnt in various approaches.

- The paper is concise and clearly presents the authors claims and results to validate these claims.

**Weaknesses:**

- The paper appears to focus on pretraining and freezing visual encoders while training policies on top of these encoders. It wasn't clear if they finetune all parameters on a given task (in the simulation and real-world policy learning experiments). In general, this may be relevant as policy performance can improve with finetuning of the entire architecture. I may have missed this in the paper but it would be good to see if the pretraining results include finetuning of the overall architecture on the robot manipulation task.

- The similarity scores for the introduced metric are quite small and rely on thresholding the binarization of the Grad-Cam output. It seems like this is a good proxy for rudimentary manipulation tasks but it doesn't necessarily seem to be a metric that will generalise to more complex settings. The metric also relies on annotations or knowledge of the task relevant features within the image.

- The pretraining dynamics alignment term incorporates chunks of state-action pairs with a history of 3 being cited as optimal. When it comes to highly dynamic tasks it is not clear how the alignment between individual images and such a history will necessarily be beneficial. For instance a single static image does not necessarily convey rich signals for dynamics (there is some signal but it is imperfect) when compared to a history of images,  aligning a state action history and a static image seems counterintuitive to me, I question whether the improvements observed are optimal relative to other methods of incorporating dynamics information. If the dynamics varied quite a lot between demonstrations on the same robot platform I question how this will effect performance. Clearly this term is working as demonstrated in the paper but I do have reservations over its usefulness and whether it is the optimal approach to incorporating dynamics information.

**Questions:**

Congratulations on this work and thanks for contributing it to the robot learning community.

I wished to clarify if during policy training does the visual encoder always remains frozen and whether you finetune the entire architecture? In the first part of the paper when evaluating your introduced metric it is clear that the encoder is frozen, I wished to confirm whether this is also the case for the reported task success rates and whether you considered finetuning the entire architecture?

---

> ### Author Response · Authors · 2024-11-22
> **Rebuttal to Reviewer cSN9**
>
> We extend our sincere thanks for your perceptive review and acknowledgment of our technical novelty, empirical contributions and presentation. We have carefully considered and responded to each of your points below.
>
> > w1 & q1:  It wasn't clear if they finetune all parameters on a given task (in the simulation and real-world policy learning experiments). In general, this may be relevant as policy performance can improve with finetuning of the entire architecture. I may have missed this in the paper but it would be good to see if the pretraining results include finetuning of the overall architecture on the robot manipulation task.
>
> Thank you for this important question. We maintain a frozen visual encoder during visuomotor control, aligning with baseline evaluation protocols. This approach offers several advantages: it ensures fair comparisons with existing methods, reduces computational costs during policy training, and effectively tests the generalization capacity of our pre-trained features. We have also conducted additional experiments with full model finetuning for all methods, where our approach maintains its superior performance advantage while showing significant improvements over the frozen encoder results reported in the manuscript. We have added these results in **Appendix A.11**.
>
> |  task  |  VC-1   | R3M  |  Ours  |
> |  ----  |  ----  | ----  | ----  |
> | can  |  82  |  19.3  |  88  |
> |  lift  |  92.7  |  92  |  99.3  |
> |  square  | 38  |  27  |  56  |
> |  aggregate  | 70.9  |  46.1  |  81.1  |
>
> > w2: It seems like the manipulation centricity is a good proxy for rudimentary manipulation tasks but it doesn't necessarily seem to be a metric that will generalise to more complex settings. The metric also relies on annotations or knowledge of the task relevant features within the image.
>
> Thank Reviewer cSN9 for this insightful suggestion. Our evaluation suite covers a broad range of complexity, as detailed in Section 2. The annotation process is quite efficient since each task needs to be annotated only once. **Leveraging SAM2, we can segment one video in approximately 1 minute, completing all required annotations in under an hour. To support the research community, we plan to release our annotated dataset for efficient pretrained model comparisons.** While we acknowledge the limitations for long-horizon tasks with complex sequential planning, utilizing language instructions with transformers for metric annotation could be a promising future direction. We have incorporated this discussion in the **limitation and future work section**.
>
> > w3: The pretraining dynamics alignment term incorporates chunks of state-action pairs with a history of 3 being cited as optimal. When it comes to highly dynamic tasks it is not clear how the alignment between individual images and such a history will necessarily be beneficial. I question whether the improvements observed are optimal relative to other methods of incorporating dynamics information. If the dynamics varied quite a lot between demonstrations on the same robot platform I question how this will effect performance. Clearly this term is working as demonstrated in the paper but I do have reservations over its usefulness and whether it is the optimal approach to incorporating dynamics information.
>
> It is a good question. We would like to clarify the dynamics alignment mechanism: The center of each state-image dynamics chunk corresponds to the same state as the static image, and with our minimum chunk length design incorporating intervening actions, this creates meaningful positive pairs for alignment. Importantly, the inserted actions serve as crucial bridges between potentially disparate states, enabling our approach to capture universal manipulation-related features through this alignment.
> While several approaches exist for pre-training robotic features, our work most closely relates to Ilija et al. [1], who explored dynamics through MAE. Our method offers a distinct and efficient approach to robotic dynamics incorporation. We acknowledge that exploring optimal dynamics integration for highly dynamic tasks is an interesting direction for future work.  We have incorporated this discussion in the **limitation and future direction section**.
>
> [1] Radosavovic et al. "Robot Learning with Sensorimotor Pre-training"
>
> We hope our explanations and additional experiments have resolved your concerns. If you have any further questions, we look forward to discussing them. Thank you again for your insightful feedback.

---

> ### Author Response · Authors · 2024-11-25
> **Does our response address your concerns?**
>
> Dear Reviewer cSN9,
>
> Thank you so much for your time and effort in reviewing our work and providing insightful feedback. As the review discussion phase concludes in two days, we kindly request you review our response and confirm if it addresses all your concerns.
>
> We welcome any further discussions and value the opportunity for continued improvement of our work.
>
> Warm regards,
>
> Paper 793 Authors

---

> > ### Comment · Reviewer_cSN9 · 2024-11-26
> > **Final Response**
> >
> > Apologies for the delay in response.
> >
> > w1 and q1 are addressed by the authors comments.
> >
> > w2 remains a major issue to the long-term relevance of this work in my opinion, however, the authors acknowledge this opinion in their updated discussion and this point of view doesn't detract from the progress that this work makes from a practical standpoint.
> >
> > w3 is appropriately addressed.
> >
> > Thank you for the rebuttal and once again apologies for the delay in responding. I will keep my scores the same as I think they fairly reflect the contribution of this work through acceptance for presentation at the conference.

---

### Official Review · Reviewer_3aRj · 2024-11-04

**Soundness:** 3
**Presentation:** 4
**Contribution:** 3
**Rating:** 8
**Confidence:** 4

**Summary:**

The paper proposes pre-train visual representations for robot manipulation by focusing on manipulation centricity, robot datasets (instead of human activity datasets), and auxiliary loss objectives for dynamic alignment and time contrastive learning. To measure Manipulation Centricity, they use Jaccard similarity between the binarized Grayscale CAM and SAM2’s foreground vs. background predictions. The proposed model RPM outperforms simulation and real robot tasks compared to baselines.

**Strengths:**

1. Detailed empirical validation across diverse tasks and simulation domains, plus real robot experiments
1. well-motivated concept of manipulation centricity with clear metrics
1. ablation studies and analysis of design choices
1. insights about benefits of robot vs human datasets for pre-training
1. Clearly written key research questions and contributions

**Weaknesses:**

The work presents insights into the visual representations that focus on manipulation centricity and dynamic alignment perform better than existing approaches. However, there are certain assumptions whose implications are not clearly discussed.
1. DROID is chosen as the robot dataset of choice instead of other larger dataset in OXE.
1. RPM is a ResNet-based model instead of directly using / comparing to pretrained models like SAM2, etc.
1. The "manipulation centricity" does not seem to improve at similar scale across different simulators, with respect to success rate.
1. A concise definition of manipulation centricity and how it is calculated should be present in the main paper, rather than existing only in appendix.

**Questions:**

1. What is the tradeoff between the visual realism of the simulator vs simulator-specific training required to achieve high success rate and manipulation centricity? For example, in RoboCasa, manipulation centricity seems really low compared to others.  How to compare it across simulators?

1. How does "manipulation centricity" depend on the choice of camera intrinsics and extrinsic with respect to the robot body frame?

1. R3M has been shown to be a poor visual representation model for robotic tasks, as compared to visual datasets [1] and pretraining image distribution matters. Why none of the baselines involve vision only representation backbones for comparison?

[1] Dasari et al, 2023. An Unbiased Look at Datasets for Visuo-Motor Pre-Training, https://data4robotics.github.io/

---

> ### Author Response · Authors · 2024-11-22
> **Rebuttal to Reviewer 3aRj**
>
> We sincerely thank Reviewer 3aRj for providing detailed and constructive feedback. Our responses to all of the concerns are addressed below. We look forward to further discussion if any points would benefit from additional clarification.
>
> > w1: 1DROID is chosen as the robot dataset of choice instead of other larger dataset in OXE.
>
> DROID stands as one of the largest robotic datasets in Open-X-Embodiment, encompassing more than 500 distinct scenes and 80 different tasks. Due to computational resource constraints (RPM training utilized only 50 hours on a single RTX 3090), we focused our experiments on the DROID dataset. **Notably, our method achieved strong generalization results across diverse robot embodiments beyond the Franka robot data provided in DROID.** While RPM can certainly be trained on other robotic datasets, our current results demonstrate its effectiveness within our computational constraints. Additionally, using the full OXE dataset would require addressing differences in action spaces across embodiments, which presents an interesting direction for future research. We have discussed it in the **limitation and future direction section.**
>
> > w2: RPM is a ResNet-based model instead of directly using / comparing to pretrained models like SAM2, etc.
>
> > q3: R3M has been shown to be a poor visual representation model for robotic tasks, as compared to visual datasets [1] and pretraining image distribution matters. Why none of the baselines involve vision only representation backbones for comparison?
>
> Thank you for this insightful suggestion. We have expanded our baseline comparisons in **Appendix A.7** including SAM2 and other pretrained models. The results below demonstrate that **RPM consistently outperforms all baselines, including both robotics-specific and vision-only approaches**:
>
> |  task  |  LIV  |  ImageNet   | SAM2  |  MAE-DROID  |  Ours  |
> |  ----  |  ----  |  ----  | ----  | ----  | ----  |
> | can  |  4.0  | 1.3  |  5.3  |  4.0  |  68.3  |
> |  lift  |  86.7  |  80.0  |  60.7  |  50.7  |  98.7  |
> |  square  | 15.3  |  4.7  |  2.0  |  6.7  | 31.3  |
> |  aggregate  | 35.3  |  28.7  |  22.7  |  20.5  |  66.1  |
>
> > w3: The "manipulation centricity" does not seem to improve at similar scale across different simulators, with respect to success rate.
>
> This is an insightful question. While manipulation centricity correlates with performance within each simulator, direct comparisons of these values across different simulators are inappropriate due to varying visual characteristics. For example, RoboCasa's high-resolution environments result in proportionally smaller target annotations, which naturally affect the absolute scale of manipulation centricity measurements.
>
> Despite these inherent differences in scales, our aggregated results demonstrate a consistent trend: **higher manipulation centricity generally corresponds to better performance within each simulator's context. This internal consistency validates the relationship between visual attention and task success.**
>
> > w4: A concise definition of manipulation centricity and how it is calculated should be present in the main paper, rather than existing only in appendix.
>
> This is a good suggestion. We have expanded **Section 3** to include detailed definitions and formulas for manipulation centricity, with Figure 3 illustrating the calculation process. The metric requires two inputs per representation for each task: a Grad-CAM video and an annotated segmentation video. Since the segmentation video is task-specific and shared across all representations, it only needs to be annotated once. We have provided the technical details for generating these inputs in **Appendix B**.

---

> > ### Author Response · Authors · 2024-11-22
> > **Rebuttal to Reviewer 3aRj**
> >
> > > q1: What is the tradeoff between the visual realism of the simulator vs simulator-specific training required to achieve high success rate and manipulation centricity?
> >
> > Thanks for raising this important point. We would like to clarify that the absolute value of manipulation centricity should not be directly used to predict downstream success rates across different simulators. This is because Grad-CAM visualizations vary significantly between simulation environments, making direct cross-simulator comparisons of manipulation centricity values inappropriate. Instead, manipulation centricity serves as a valuable comparative metric for ranking different representations within the same domain or aggregated domains. We have highlighted this crucial distinction in **Figure 2** of our manuscript.
> >
> > > q2: How does "manipulation centricity" depend on the choice of camera intrinsics and extrinsic with respect to the robot body frame?
> >
> > The computation of manipulation centricity does not directly depend on camera intrinsics or extrinsics since it is derived from Grad-CAM videos and annotated segmentation videos, both of which are independent of specific camera parameters. Additionally, our evaluation suite deliberately includes diverse camera configurations to ensure robustness across different setups.
> >
> > While the absolute manipulation centricity values may vary with changes in camera intrinsics or extrinsics, the overall positive correlation between manipulation centricity and task success remains consistent. To demonstrate this, we conducted additional experiments using Robomimic demonstrations captured from an alternative camera viewpoint (eye-in-hand). Policies trained and evaluated under this configuration showed consistent trends:
> >
> > |  method  |  manipulation centricity   | success rate  |
> > |  ----  |  ----  | ----  |
> > |  VC-1  |  0.071  |  56.9  |
> > |  R3M  | 0.069  |  55.8  |
> > |  Ours  | 0.076  |  61.8  |
> >
> > **These results demonstrate that our metric maintains its positive correlation with task success, even though the absolute values of manipulation centricity may vary with camera parameters.** We have included this detailed analysis in **Appendix A.10.**
> >
> > In conclusion, we have addressed all reviewer questions and improved our paper following these insightful comments. We have sharpened our technical definitions, expanded our experiments on more baselines, and clarified our evaluation protocols. We welcome further discussion if needed.

---

> > > ### Author Response · Authors · 2024-11-25
> > > **Does our response address your concerns?**
> > >
> > > Dear Reviewer 3aRj,
> > >
> > > Thank you so much for your time and effort in reviewing our work and providing insightful feedback. As the discussion phase concludes in 48 hours, we kindly request you review our response and confirm if it addresses all your concerns.
> > >
> > > We welcome any further discussions and value the opportunity for continued improvement of our work.
> > >
> > > Best regards,
> > >
> > > Paper 793 Authors

---

> > > > ### Comment · Reviewer_3aRj · 2024-11-26
> > > > **Reviewed rebuttal; Updated score**
> > > >
> > > > Dear authors, I have carefully reviewed your responses. Thank you for additional experiments with much larger vision only models.
> > > > Two minor clarifications requested:
> > > >
> > > > (a) "RoboCasa's high-resolution environments result in proportionally smaller target annotations, which naturally affect the absolute scale of manipulation centricity measurements." - What does this mean? What is the connection between simulator's high-resolution, smaller target annotations and its impact of "manipulation centricity" calculation?
> > > >
> > > > (b) Can you explain the symbols in Equation 1? What do you mean by "F_{SAM2}(I), Grad"?
> > > >
> > > > Overall, I understand that manipulation centricity as a metric maintains only a correlation with task success, and its absolute values may be affected by the resolution of the image and relative proportion of pixels showing the interaction object vs background.

---

> > > > > ### Author Response · Authors · 2024-11-27
> > > > > **Response to Reviewer 3aRj**
> > > > >
> > > > > Dear Reviewer 3aRj,
> > > > >
> > > > > Thank you for your insightful questions. Please find our detailed responses below:
> > > > >
> > > > > > What is the connection between simulator's high-resolution, smaller target annotations and its impact of "manipulation centricity" calculation?
> > > > >
> > > > > In high-resolution simulators like RoboCasa, while objects are rendered with more detail (more pixels), they actually occupy a smaller proportion of the total image space compared to lower-resolution simulators. For example, the same physical object might take up 5% of the total pixels in a high-resolution environment but 15% in a lower-resolution one.
> > > > > This affects manipulation centricity calculations because they rely on proportional measurements.
> > > > >
> > > > > > Can you explain the symbols in Equation 1? What do you mean by "F_{SAM2}(I), Grad"?
> > > > >
> > > > > Thank you for bringing this to our attention. There was indeed a typo error in the Grad-CAM equation. To clarify:
> > > > >
> > > > > * $F_{SAM2}(I)$ denotes the ground truth segmentation masks predictes by SAM2
> > > > > * $\mathrm{Grad}\text{-}\mathrm{CAM}(\mathcal{F}_{\phi}(I)\big)$ presents the binarized Grad-CAM outputs of the encoded image.
> > > > >
> > > > > We have corrected this typo in our revised manuscript.
> > > > >
> > > > > We sincerely appreciate your careful review and attention to detail, which has helped us improve the clarity of our work. If you have any further questions or concerns, we look forward to discuss with you.
> > > > >
> > > > > Best regards,
> > > > >
> > > > > Paper 793 Authors

---

### Official Review · Reviewer_wcpD · 2024-11-07

**Soundness:** 3
**Presentation:** 4
**Contribution:** 3
**Rating:** 8
**Confidence:** 5

**Summary:**

The paper first proposes manipulation centricity, a new metric for evaluating representations for robotics, and finds that it is a strong indicator of success rates in downstream robotic manipulation tasks. Building on this insight, it proposes RPM, a framework for learning visual representation for robotics from large-scale robotics dataset. Experiments in both simulation and the real world show that RPM leads to improved manipulation centricity and higher task success rate when evaluated using behavior cloning.

**Strengths:**

- The paper is clear and well presented
- The simulation experiments are rigorous and have a lot of diversity
- The paper presents a novel metric for evaluating visual representations for robotics, and novel findings on how to integrate robotics information (states, actions) for training better representations

**Weaknesses:**

- There could be more empirical analysis and intuition regarding how each loss affects manipulation centricity. Authors could explain intuitively how each loss contributes to better MC and visualize the gradcam of an encoder trained with each loss
- The paper lacks comparison to more state-of-the-art baselines, such as VIP/LIV, or object-centric pre-trained representations such as POCR/HODOR
- As I understand, the goal here is to make the representation pay attention to task-relevant regions e.g. robot and task-relevant objects. The authors have already demonstrated that this information can be obtained using SAM2. Have the authors considered directly using the masks as a form of supervision for training the representation? I think this could serve as a baseline, since RPM requires additional priveleged information that only a robotics dataset has, i.e. state and action, whereas we can generate segmentation masks for any general sources of data.
- The paper could benefit from more extensive real robot experiments

**Questions:**

- Why is it still necessary to attend to the manipulator region when there is already proprioception input?
- What is the evaluation protocol in the real world? How are the environments varied between training and evaluation, and what would count as a success?
- By attending the robot and task-relevant object, technically this would allow the representation to have stronger generalization capabilities. Perhaps the author should run some visual generalization experiments to find out if RPM enables better generalization.

---

> ### Author Response · Authors · 2024-11-22
> **Rebuttal to Reviewer wcpD**
>
> We greatly appreciate the reviewer's thoughtful feedback. We have addressed each point and comment raised in the review below. Should any aspects require further clarification, we welcome continued discussion.
>
> > w1: There could be more empirical analysis and intuition regarding how each loss affects manipulation centricity. Authors could explain intuitively how each loss contributes to better MC and visualize the gradcam of an encoder trained with each loss
>
> We have expanded the qualitative and quantitive analysis in **Appendix A.9** to illustrate how each loss influences manipulation centricity. Our analysis reveals that each loss serves a distinct purpose: The dynamics alignment loss guides the encoder to capture rich manipulation-related features by focusing on the most dynamic elements of the scene. The action prediction loss directly enhances the model's action prediction capabilities by leveraging the pre-trained features. The time contrastive loss facilitates the learning of temporal visual relationships. **Our experimental results confirm that all three losses contribute meaningfully to both improved manipulation centricity and higher downstream task success rates.**
>
> Moreover, we would like to emphasize two other key conclusions: (1) the dynamics alignment loss provides the most substantial benefit to representation learning, as it significantly enhances the encoder’s focus on manipulation-centric features; and (2) the strong positive correlation between manipulation-centricity and downstream task success rates still hold, strengthening the key contribution of this paper. Thanks for the good suggestion!
>
> |  method  |  manipulation centricity   | success rate  |
> |  ----  |  ----  | ----  |
> | Ours  |  0.1752  |  83.2  |
> |  Ours w/o dynamics alignment  |  0.1001  |  66.2  |
> |  Ours w/o action prediction  | 0.1426  |  71.3  |
> |  Ours w/o temporal contrast   | 0.1454  |  72.0  |
>
> > w2: The paper lacks comparison to more state-of-the-art baselines, such as VIP/LIV, or object-centric pre-trained representations such as POCR/HODOR.
>
> We appreciate these constructive suggestions. Regarding baseline selection, we note that VC-1 (NeurIPS 2023) has already demonstrated its superiority over VIP, while both VC-1 and HRP (RSS 2024) represent the current state-of-the-art in robotic representation learning. While HODOR addresses tasks outside the robotics domain and POCR is not open-sourced, **we have added discussions of these relevant methods in the related work section in the manuscript**. Additionally, we have expanded our evaluation to include comparisons with LIV, ImageNet, SAM2 and MAE trained on DROID on Robomimic (Appendix A.7). **The results below demonstrate that our method significantly outperforms all baselines.**
>
> |  task  |  LIV  |  ImageNet   | SAM2  |  MAE-DROID  |  Ours  |
> |  ----  |  ----  |  ----  | ----  | ----  | ----  |
> | can  |  4.0  | 1.3  |  5.3  |  4.0  |  68.3  |
> |  lift  |  86.7  |  80.0  |  60.7  |  50.7  |  98.7  |
> |  square  | 15.3  |  4.7  |  2.0  |  6.7  | 31.3  |
> |  aggregate  | 35.3  |  28.7  |  22.7  |  20.5  |  66.1  |

---

> ### Author Response · Authors · 2024-11-22
> **Rebuttal to Reviewer wcpD**
>
> > w3: Have the authors considered directly using the masks as a form of supervision for training the representation?
>
> This is a good question. While utilizing a dataset with segmentation masks for manipulation-related regions would be an interesting direction to explore, it is currently impractical due to the substantial annotation costs associated with labeling our large-scale robotic dataset. Moreover, segmentation-based supervision alone may not sufficiently capture the complex state-action dynamics that are critical for learning effective and generalizable representations, as demonstrated by our approach. We have incorporated this discussion in the **limitation and future direction section**.
>
> > w4: The paper could benefit from more extensive real robot experiments
>
> We completely agree that additional real-world evaluations, specifically extending to more complex tasks with various objects such as Fold Clothes and Pick Place AAA Battery, would further strengthen our work.  We have incorporated this discussion in the limitation and future direction section. We are currently collecting additional expert demonstrations for these extended tasks, as open-source demonstrations are insufficient for imitation learning. While resource and time constraints limit our ability to show more real-world results during the discussion period, we plan to expand these evaluations in the camera-ready version. Meanwhile, we emphasize that our current three tasks already demonstrate considerable complexity across diverse objects and manipulation skills, effectively validating our method's capabilities.
>
> > q1: Why is it still necessary to attend to the manipulator region when there is already proprioception input?
>
> This is an insightful question. Visual information of the manipulator region provides richer contextual cues that proprioceptive input alone cannot capture, including the relative spatial relationships between the end effector and objects, as well as the manipulator's approximate pose in the world frame. By explicitly attending to the manipulator during pre-training, our encoder develops an enhanced capacity to extract critical manipulation-relevant features, making this approach both distinctive and advantageous.

---

> > ### Author Response · Authors · 2024-11-22
> > **Rebuttal to Reviewer wcpD**
> >
> > > q2: What is the evaluation protocol in the real world? How are the environments varied between training and evaluation, and what would count as a success?
> >
> > The real-world evaluation protocol is detailed in Section 5.2 of our paper. To ensure a fair comparison, we evaluate all methods under identical starting conditions that were not present in the demonstration data. While our training data was collected within a confined region, our evaluation protocol introduces significant variations.
> >
> > For each task, we maintain specific success criteria:
> >
> > Lift: The robot arm starts from a fixed initial pose, with a sandbag randomly positioned on a fixed base. Success requires lifting the sandbag at least 10 cm above the base.
> > Sweep: The dustpan, trash, besom, and rack positions are randomly initialized. Success is achieved when the trash is successfully swept into the dustpan using the besom.
> > Rearrange: The pot and surrounding objects are randomly positioned. Success requires placing the pot precisely on the designated black circular area of the cooking range.
> >
> > We have incorporated these detailed evaluation protocols into **Appendix A.1**. Importantly, this standardized evaluation framework is applied consistently across all methods, ensuring fair comparison of their performance in real-world settings.
> >
> > > q3: By attending the robot and task-relevant object, technically this would allow the representation to have stronger generalization capabilities. Perhaps the author should run some visual generalization experiments to find out if RPM enables better generalization.
> >
> > Thanks for the good suggestion! We compare the generalization ability of some methods on Robomimic. While maintaining our original training protocol, we introduced color perturbations during policy evaluation by applying ColorJitter transformations to the visual observations.
> >
> > Specifically, our evaluation protocol involved selecting optimal model checkpoints from the training process, conducting 20 evaluation episodes (each with a duration of 200 timesteps), and recording success rates under standard conditions. We then repeated the evaluation with added visual perturbations using 'torchvision.transforms.ColorJitter(brightness=0.1, hue=0.1)'. The comparative results (has been included in **Appendix A.8**) demonstrate that **our method exhibits superior robustness to visual perturbations, indicating stronger generalization capabilities across varying visual conditions.**
> >
> > |  method  |  original performance   | performance after ColorJitter  |  performance drop ratio  |
> > |  ----  |  ----  | ----  | ----  |
> > |  ours  | 95  | 83 | 12.63% |
> > |  vc-1  | 83  | 56.7 | 31.69% |
> > |  r3m  | 86.7  | 25 | 71.16% |
> >
> > In summary, we have detailedly responded to all reviewer concerns and enhanced our manuscript based on these valuable suggestions, specifically through improved evaluation protocols, baseline comparisons, and ablation analyses. We appreciate the insightful feedback that has strengthened our paper's clarity and technical depth, and welcome any further discussion if needed.

---

> > > ### Author Response · Authors · 2024-11-25
> > > **Does our response address your concerns?**
> > >
> > > Dear Reviewer wcpD,
> > >
> > > Thank you so much for your time and effort in reviewing our work and providing insightful feedback. As the discussion phase concludes in 48 hours, we kindly request you review our response and confirm if it addresses all your concerns.
> > >
> > > We welcome any further discussions and value the opportunity for continued improvement of our work.
> > >
> > > Best regards,
> > >
> > > Paper 793 Authors

---

> > > > ### Comment · Reviewer_wcpD · 2024-11-28
> > > > **Response to Rebuttal**
> > > >
> > > > Apologies for the late response.
> > > >
> > > > All of my concerns and questions are addressed by the authors, and I think some of the new results have led to interesting new conclusions. Great effort and congrats! I have raised my score.
> > > >
> > > > Regarding w2, HODOR is related to robotics. The POCR/HODOR line of work proposes pre-trained visual representations for robotics that are object centric. Since they explicitly segment out the task-relevant object and the robot and use them as input to the policy, I strongly believe they will result in better MC and task success rates. I do understand that they haven't released code yet, but if the code is released before camera ready, I think the authors should run the experiments and add them as baselines.
> > > >
> > > > Regarding q3, strong generalization is pretty cool! The authors should consider highlighting this more in the final version, e.g. by adding other types of visual distribution shift. Incidentally, prior work (https://kayburns.github.io/segmentingfeatures/) has shown that emerging segmentation capability is a strong indicator of the generalization ability of a visual representation. Could there be some correlation between MC and segmentation capability? May be worthwhile to investigate.

---

> > > > > ### Author Response · Authors · 2024-11-28
> > > > > **Response to Reviewer wcpD**
> > > > >
> > > > > Dear Reviewer wcpD,
> > > > >
> > > > > Thank you again for your constructive feedback on our response which will help strengthen the final version of our paper.
> > > > >
> > > > > * We appreciate your suggestion regarding object-centric approaches. We will actively monitor the release of their code and commit to including comparative experiments in our camera-ready version if available.
> > > > >
> > > > > * We strongly agree that the generalization analysis deserves more emphasis in the main text. Following your suggestion, we plan to expand this section in the camera-ready version by:
> > > > > 1. Including additional visual distribution shift scenarios to further demonstrate RPM's generalization capability
> > > > > 2. Providing a more detailed analysis of the results
> > > > >
> > > > > * In our related work section, we have discussed Burns et al.'s work on emerging segmentation as a predictor of representation quality. Your suggestion is very insightful. We will examine potential correlations between MC scores and emergent segmentation patterns and include these new insights in the appendix.
> > > > >
> > > > > We are sincerely grateful for the time and effort you dedicated to the discussion period.
> > > > >
> > > > > Best regards,
> > > > >
> > > > > Paper 793 Authors

---

### Author Response · Authors · 2024-11-22
**General Response**

## Review Highlight

We sincerely thank all reviewers for their thorough feedback and constructive suggestions.
We appreciate the positive comments from all reviewers on our paper's clarity and presentation. In particular, we are grateful for the acknowledge of:

* A novel manipulation centricity metric (**Reviewers 3aRj, cSN9, vJK6**)
* Innovative integration of robotics information in representation learning (**Reviewers wcpD, vJK6**)
* Comprehensive validation across simulated and real-robot tasks (**Reviewers wcpD, 3aRj, cSN9**)

The reviewers' feedback has helped strengthen our work significantly. We have addressed their questions and suggestions thoroughly in our responses.

## Reviewer concerns

We have taken the reviewers' feedback into account and responded to each question with detailed explanations and additional experiment results. For the main concerns:
* Concerns about Manipulation Centricity (Reviewers wcpD, 3aRj, cSN9, vJK6): Enhanced quantitative analysis of each component's effects on manipulation centricity, with technical details, definitions, and evaluation protocol descriptions added.
* Concerns about Empirical Evaluation (Reviewers wcpD, 3aRj, vJK6): Clarified our evaluation on 23 diverse manipulation tasks and added four new baselines (LIV[1], ImageNet[2], SAM2[3], MAE-DROID[4]), demonstrating superior performance in both simulation and real-world settings.
* Concerns about Generalization (Reviewers 3aRj, wcpD, cSN9): Validated robustness under color perturbations, varying camera viewpoints and additional finetuning experiments, while demonstrating strong generalization.
* Concerns about Technical Novelty (Reviewer vJK6): Highlighted two key contributions: systematic analysis of manipulation-centricity impact on downstream tasks, and first incorporation of state-action alignment and action prediction in robotic representation pretaining.

These responses address key concerns while providing new experimental results that strengthen our paper's contributions. We incorporate our clarifications and additional analyses in our revised manuscript.

[1] Ma et al. "LIV: Language-Image Representations and Rewards for Robotic Control"

[2] Dasari et al, 2023. "An Unbiased Look at Datasets for Visuo-Motor Pre-Training"

[3] Ravi et al. "SAM 2: Segment Anything in Images and Videos"

[4] He et al. "Masked Autoencoders are Scalable Vision Learners"

---

> ### Author Response · Authors · 2024-11-22
> **Summary of paper updates**
>
> We have enhanced our manuscript in several key aspects:
>
> Main Text:
> * Added detailed definitions and formulas for manipulation centricity (Section 3)
> * Enhanced related work discussion of recent representation methods (Section 6)
> * Added overview figure illustrating our methodology and metric (Figure 1)
> * Clarified cross-simulator evaluation protocols (Figure 2)
>
> Appendix:
> * Expanded comparisons with 4 new baselines(A.7)
> * Analysis of visual generalization under perturbations (A.8)
> * Comprehensive ablation studies on loss components (A.9)
> * Study of camera parameter effects on manipulation centricity (A.10)
> * Additional finetuning experiments and results (A.11)
> * Quantitative analysis of manipulation centricity for Table 1 tasks (A.12)
> * Discussion of limitation and future directions (Section LIMITATION AND FUTURE DIRECTIONS)
>
> These updates are temporarily highlighted in "blue" for your convenience to check.
> We strongly believe that our paper makes a great contribution to the ICLR robotics community, particularly because reviewers’ constructive comments enhanced the manuscript.
>
> We sincerely thank the reviewers and the AC for their time and thoughtful feedback on our paper during the discussion period. We hope that our responses have effectively addressed all the questions and concerns, and eagerly await further discussion and feedback.

---

### Author Response · Authors · 2024-12-02
**A letter to AC: Thanks for organizing and supervising the review process and concerns about Reviewer vJK6**

Dear Area Chair and Senior Area Chair,

We sincerely appreciate your time in considering this message and express our gratitude for your dedication to organizing and supervising the ICLR review process.

During the discussion period, we received positive feedback from three reviewers (**wcpD, 3aRj, and cSN9**), all providing a Rating of 8 with further discussions. These reviewers strongly acknowledged our paper's contributions and recommending its acceptance to ICLR.

Despite multiple reminders, Reviewer **vJK6** has not participated during the discussion period. Given the lack of engagement and divergent opinions from other reviewers, we are concerned about potential bias in the assessment.

For Reviewer **vJK6**'s concerns, our response:
* Provided detailed clarifications about our technical contributions (w1 & w2). These aspects of our work were specifically highlighted by the other reviewers as demonstrating strong technical novelty and thorough experimental validation.
* Addressed Reviewer vJK6's writing-related concerns (w3 & w6) with additional explanations to better convey our motivation, which have been incorporated into our updated manuscript.
* Included new ablation experiments to address the misunderstandings of our previous results (w4 & w5). As presented in our general response, we added six additional experimental studies, which received positive acknowledgment from the other reviewers.

In summary, we have thoroughly addressed all concerns raised by Reviewer **vJK6**, though unfortunately received no response. We hope this information proves helpful during the AC-reviewer discussion period.

We are grateful for your consideration and thank AC and all reviewers for their valuable contributions to the community.

Warm Regards,

Paper 793 Authors

---

### Meta-Review · Area_Chair_4PyR · 2024-12-21

**Metareview:**

The paper introduces a novel metric called “manipulation-centricity” (MC) for evaluating pretrained representations and demonstrates its correlation with downstream manipulation task performance. Building on these insights, it proposes RPM, a representation learning method that captures both visual features and dynamic information to enhance manipulation-centricity (MC). Through extensive experiments, the authors validate that their approach improves task success rates compared to baseline methods.

Strengths:
- Novel Metric and Concept: “Manipulation-centricity” as a quantitative metric for evaluating visual representations in robotic manipulation is innovative and well-motivated.
- Experiments: The paper provides rigorous validation across diverse tasks and simulators, demonstrating the robustness of RPM.
- Clear Writing and Structure: The paper is well-organized and presents its key ideas well.

Weaknesses
- Real-World Experiments: While the paper includes real-world evaluations, the scope of these experiments is limited. Conducting more complex tasks with diverse objects could strengthen the results.
- Missing Baseline Comparisons: As note by Reviewer wcpD, the POCR/HODOR line of work proposes pretrained visual representations for robotics that are object-centric. These could serve a strong basline for the current method. The authors have noted that the code for HODOR is not yet available; if it is released before the camera-ready deadline, I recommend running these experiments and including them as baselines.

The paper demonstrates significant novelty in proposing manipulation-centricity as a metric and aligning visual representations with robotic dynamics. It provides thorough experimental validation and offers meaningful insights into pretraining for robotic manipulation tasks. However, the limited scope of real-world experiments slightly diminishes the overall impact.

Despite these limitations, the strengths of the paper, particularly its novel contributions and rigorous empirical analysis, outweigh the weaknesses. Therefore, I recommend acceptance, with the suggestion to address the noted weaknesses in the camera-ready version.

**Additional Comments On Reviewer Discussion:**

During the rebuttal period, the authors addressed the key points raised by the reviewers. However, some concerns noted in the weaknesses section remain, which the authors have promised to address in the next version of the paper.

One of the reviewers had a more negative stance on the paper and did not engage with the authors. I believe the authors have adequately addressed this reviewer's concerns. Therefore, I recommend accepting the paper.

---

### Decision · Program_Chairs · 2025-01-22

Accept (Poster)